



# Rate–induced tipping cascades arising from interactions between the Greenland Ice Sheet and the Atlantic Meridional Overturning Circulation

Ann Kristin Klose[1,2], Jonathan F. Donges[1,3], Ulrike Feudel[4], and Ricarda Winkelmann[1,2]

[1]FutureLab Earth Resilience in the Anthropocene, Earth System Analysis & Complexity Science, Potsdam Institute for Climate Impact Research (PIK), Member of the Leibniz Association, P.O. Box 6012 03, D-14412 Potsdam Germany
[2]Institute of Physics and Astronomy, University of Potsdam, 14476 Potsdam, Germany
[3]Stockholm Resilience Centre, Stockholm University, Stockholm, SE-10691, Sweden
[4]Theoretical Physics/Complex Systems, ICBM, University of Oldenburg, 26129 Oldenburg, Germany

**Correspondence:** Ann Kristin Klose (annkristin.klose@pik-potsdam.de) and Ricarda Winkelmann (ricarda.winkelmann@pik-potsdam.de)

**Abstract.** The Greenland Ice Sheet and Atlantic Meridional Overturning Circulation are considered tipping elements in the climate system, where global warming exceeding critical threshold levels in forcing can lead to large–scale and nonlinear reductions in ice volume and overturning strength, respectively. The positive–negative feedback loop governing their interaction (with a destabilizing effect on the AMOC due to ice loss and subsequent freshwater flux into the North Atlantic as well as a

stabilizing effect of a net–cooling around Greenland with an AMOC weakening) may determine the long–term stability of both tipping elements. Here we explore the potential dynamic regimes arising from this positive–negative tipping feedback loop in a process–based conceptual model. Under idealized forcing scenarios we identify conditions under which different kinds of tipping cascades can occur: Herein, we distinguish between overshoot tipping cascades (leading to tipping of both GIS and AMOC) and rate–induced tipping cascades (where the AMOC despite not having crossed its own intrinsic tipping point tips

nonetheless due to the fast rate of ice loss from Greenland). These different cascades occur within corridors of distinct tipping pathways that are affected by the GIS melting patterns and thus eventually by the imposed forcing and its time scales. Our results suggest that it is not only necessary to avoid breaching the respective critical levels of the environmental drivers for the Greenland Ice Sheet and Atlantic Meridional Overturning Circulation, but also to respect *safe rates* of environmental change to mitigate potential domino effects.

## 1   Introduction

The Greenland Ice Sheet (GIS) and the Atlantic Meridional Overturning Circulation (AMOC) have been identified as possible interacting tipping elements of the climate system, transitioning into a qualitatively different state once a critical threshold in forcing levels of their respective environmental drivers is crossed (Lenton et al., 2008; Armstrong McKay et al., 2022).

Both components of the Earth system may be propelled towards an alternative state by positive feedback mechanisms (such

as the melt–elevation feedback in Greenland or the salt–advection feedback relevant for AMOC dynamics) with the crossing





of a tipping point (Levermann et al., 2012). From a mathematical viewpoint, different mechanisms for critical transitions have been identified (Ashwin et al., 2012; Halekotte and Feudel, 2020). Tipping towards a qualitatively different state may be induced when a bifurcation point is transgressed by a slowly changing control parameter of the system (bifurcation–induced tipping) (Ashwin et al., 2012). By contrast, a system in its bistable regime may be driven to its alternative state by noise without

a change in external conditions (noise–induced tipping) (Ashwin et al., 2012; Ditlevsen and Johnsen, 2010). Moreover, a system can be pushed into another state by one singular shock perturbation or extreme event (shock tipping) (Halekotte and Feudel, 2020; Schoenmakers and Feudel, 2021). Finally, a transition to a different system state due to a control parameter change exceeding a critical rate at which the system fails to track its changing quasi–steady equilibrium is called rate–induced tipping (Wieczorek et al., 2011; Ashwin et al., 2012; Vanselow et al., 2019; Lohmann and Ditlevsen, 2021).

The Greenland Ice Sheet and Atlantic Meridional Overturning Circulation are strongly linked via freshwater fluxes into the North Atlantic originating from a melting GIS on the one hand, and via a relative cooling around Greenland with a slowdown of the AMOC on the other hand (Kriegler et al., 2009; Sinet et al., 2023). More specifically, the increasing mass loss of the GIS (Shepherd et al., 2020; Mouginot et al., 2019; Van den Broeke et al., 2016) results in a freshwater input to the North Atlantic (Bamber et al., 2012, 2018; Trusel et al., 2018), which is assumed to weaken the AMOC by decreasing sea water density and

thereby weakening deep water formation (Caesar et al., 2018; Rahmstorf et al., 2015). The weakening or even tipping of the AMOC may be accompanied by a reduced northward heat transport and thus a relative cooling around Greenland (Vellinga and Wood, 2002, 2008; Jackson et al., 2015), which, in turn, may act in a stabilizing way on the melting processes of the GIS (Kriegler et al., 2009). The effect of this positive–negative feedback loop on the overall stability of the coupled system of climatic tipping elements is largely unknown.

The potential for cascades arising from tipping element interactions such as the feedback loop between the ice sheet on Greenland and the Atlantic Meridional Overturning Circulation has been addressed by modelling efforts of different complexity. Building on Abraham et al. (1991) and Brummitt et al. (2015), the qualitatively different dynamics arising from interactions of idealized tipping elements and preconditions for the emergence of tipping cascades have been studied (Dekker et al., 2018; Klose et al., 2020, 2021). The propagation of tipping cascades on complex networks is affected by the network topology with

clustering and spatial organization increasing the susceptibility to cascades (Krönke et al., 2020). In particular, small–scale motifs promote tipping cascades by decreasing the critical coupling strength to trigger a tipping cascade (Wunderling et al., 2020b).

Within the climate system, interactions between several large–scale tipping elements including the AMOC and the Greenland Ice Sheet as well as the West Antarctic Ice Sheet and the Amazon rainforest have been described (Kriegler et al., 2009;

Gaucherel and Moron, 2017) and the arising dynamics may involve cascades (Lenton et al., 2019; Rocha et al., 2018). The interactions between these four key climate tipping elements tend to be overall destabilizing under ongoing warming as found by integrating expert knowledge and including uncertainties of critical temperature thresholds and interaction strengths into a risk analysis approach for these interacting tipping elements (Wunderling et al., 2023, 2021, 2020a). Employing physical process–based but still conceptual models, it was demonstrated that the intensification of ENSO, which is associated with

growing oscillations of eastern Pacific sea surface temperatures after the crossing of a Hopf bifurcation, may be initiated by





an AMOC collapse (Dekker et al., 2018). The AMOC may, in turn, be stabilized by a disintegration of the West Antarctic Ice Sheet, thereby potentially hindering cascading tipping in the climate system (Sinet et al., 2023).

Significant changes of both systems with an acceleration of GIS mass loss (Shepherd et al., 2020; Trusel et al., 2018) as well as a weakening of the AMOC (Caesar et al., 2018) imply the approach of a critical threshold with ongoing global warming

(Boers and Rypdal, 2021; Boers, 2021). In addition, triggering and transmission of abrupt changes of these systems by ice–ocean interactions may have occurred in the past as suggested by paleoevidence (Brovkin et al., 2021; Thomas et al., 2020). Guided by present–day observations and insights from paleoclimate records, the potential future dynamics of the coupled GIS–AMOC system have been explored in the framework of e.g. hosing experiments (compare Sect. 2 for further details). However, the effects of a possible non–linear disintegration of the GIS with different rates and the additional negative feedback

via temperature changes around Greenland for cascading tipping behaviour have not been explicitly considered on long time scales yet.

Here, we qualitatively explore the dynamics and in particular the risk of cascading tipping behaviour emerging from the interaction of GIS and AMOC via a positive–negative feedback loop of freshwater fluxes into the North Atlantic and a relative cooling around Greenland. In Sect. 2 we give more details on changes observed at present, constraints from paleoclimate

evidence for the potential future behaviour and previous modelling approaches of the coupled GIS–AMOC system, which motivate our study. The interaction of the GIS and AMOC is captured by coupled process–based while still conceptual models of both climatic tipping elements (Wood et al., 2019; Levermann and Winkelmann, 2016) (Sect. 3 and Sect. 4.1). The aim here is not to provide quantitative statements or projections on the emergence of tipping cascades in the climate system. Rather, our approach allows us to examine the qualitative behaviour of the coupled system under a multitude of forcing scenarios and on

long time scales. Complementing freshwater hosing experiments, we study the AMOC response to a decline of the GIS under idealized forcing scenarios yielding distinct GIS melting patterns (Sect. 4.2). These include a rate–induced cascade where the AMOC tips due to the rapid ice loss from Greenland without having crossed its own tipping point yet. To this end, we show that the potentially stabilizing effect of the relative cooling around Greenland due to an AMOC slowdown may prevent a tipping of the GIS only conditionally for a limited forcing given that the AMOC resides close to its threshold (Sect. 4.3). These findings

are relevant for defining safe pathways of environmental change to maintain the resilience of the Earth system (Sect. 5).

## 2 Greenland Ice Sheet and Atlantic Meridional Overturning Circulation as interacting tipping elements

Here, we explore current observations on the state of the individual tipping elements as well as paleoevidence for past tipping cascades in more detail. These insights form the basis for assessing the future stability of the interacting Greenland Ice Sheet and the AMOC under ongoing global warming. Previous modelling approaches capturing aspects of the coupled GIS–AMOC

system and determining potentially arising dynamics are presented and their limitations are discussed.

**Observed changes** Observations reveal pronounced changes of both systems: At present, the Greenland Ice Sheet is losing mass at an accelerating rate due to an increase in surface melt and ice discharge (Shepherd et al., 2020; King et al., 2020), to-





taling to a loss of 3902±342 Gt of ice between 1992 and 2018 (Slater et al., 2021). The AMOC may have reached its weakest
state in at least a millennium (Caesar et al., 2021) after a slowdown in the past decades (Rahmstorf et al., 2015; Caesar et al.,
2018). Early warning signals indicate the proximity of a critical threshold in west Greenland (Boers and Rypdal, 2021) and a
loss of stability of the current strong AMOC mode (Boers, 2021).

**Paleoevidence of tipping interactions** In Earth history, strong retreats of the Greenland Ice Sheet (e.g., during the Pliocene
and interglacials of the Pleistocene; Dutton et al., 2015; Schaefer et al., 2016; Christ et al., 2021) and a slowdown of the
AMOC (e.g., during the last glacial period; Rahmstorf, 2002; Ritz et al., 2013; Lynch-Stieglitz, 2017) have likely occurred.
Paleoclimate evidence suggests that some abrupt changes of the AMOC and the Greenland Ice Sheet may have been mediated
by cryosphere–ocean interactions (Brovkin et al., 2021; Thomas et al., 2020). Large regional temperature changes in Green-
land during the last glacial period are associated with changes of the AMOC (Lynch-Stieglitz, 2017; Barker and Knorr, 2016).
In turn, past AMOC regime shifts are connected to freshwater pulses into the North Atlantic originating from a changing
cryosphere (compare Brovkin et al., 2021).

**Previous modelling approaches** The fate of the AMOC in response to a freshwater flux from Greenland, i.e. the effects of
a unidirectional coupling of the GIS towards the AMOC, was studied in terms of freshwater hosing experiments in General
Circulation Models (GCMs) (Hu et al., 2009; Jungclaus et al., 2006; Stouffer et al., 2006; Swingedouw et al., 2013, 2015;
Rahmstorf, 1995). In addition, experiments with coupled climate–ice sheet models under global warming were conducted
(Driesschaert et al., 2007; Fichefet et al., 2003; Gierz et al., 2015; Golledge et al., 2019; Mikolajewicz et al., 2007; Ridley
et al., 2005; Swingedouw et al., 2006; Winguth et al., 2005). In general, the AMOC response to a freshwater flux associated
with a GIS melting ranges from no significant weakening to an observable effect on the AMOC strength (Driesschaert et al.,
2007; Fichefet et al., 2003; Gierz et al., 2015; Golledge et al., 2019; Hu et al., 2009; Jungclaus et al., 2006; Mikolajewicz et al.,
2007; Ridley et al., 2005; Swingedouw et al., 2006; Winguth et al., 2005). A collapse of the AMOC was found by Stouffer et al.
(2006) in response to a freshwater input of 1.0 Sv for 100 years and by Fichefet et al. (2003) in simulations of the 21st century
climate. The AMOC trajectory under temporary freshwater input depends among others on the sensitivity of the considered
model and the background climate state (Swingedouw et al., 2013, 2015). However, freshwater inputs into the North Atlantic
in such hosing experiments are highly idealized, vary in terms of their magnitude as well as spatial and temporal characteristics
and do not take into account the nonlinear melting characteristics of a tipping of the ice sheet on Greenland (Trusel et al., 2018;
Robinson et al., 2012). In addition, the potential stabilizing effect of relatively colder temperatures in Greenland on the ice
sheet (Jackson et al., 2015) is not included. Many Earth system models are debated to be biased towards a too stable AMOC
and hence may not be able to resolve its nonlinear behaviour due to missing couplings, processes and feedbacks, uncertainties
in their representation and biases in fluxes of salt and heat between ocean basins (Liu et al., 2017; Valdes, 2011; Weijer et al.,
2019). Finally, computational constraints impede assessing multiple potential AMOC trajectories under uncertain parameters
and climate forcings on long time scales (Wood et al., 2019; Jackson and Wood, 2018). However, considerations on long time
scales are relevant given the rather slow ice sheet response to perturbations in its climatic boundary conditions but also to





determine the state to which the AMOC eventually converges after a freshwater perturbation (Fichefet et al., 2003; Jackson
and Wood, 2018; Weijer et al., 2019). The hosing experiments were supplemented by more conceptual approaches allowing
for an uncertainty analysis of the future development of the AMOC overturning strength under global warming and ice sheet
melting (Zickfeld et al., 2004; Bakker et al., 2016).

Recently, a possible rate–induced tipping (Ashwin et al., 2012) of the AMOC for a quickly changing, time–dependent
freshwater forcing in a three–dimensional ocean model (Lohmann and Ditlevsen, 2021) confirmed the suggested sensitivity
of the AMOC to the rate of driver change (Stocker and Schmittner, 1997; Alkhayuon et al., 2019). It may further hint to
cascading tipping of the interacting GIS and AMOC due to time scale differences between e.g. the freshwater input and the
AMOC response time scale (compare e.g. Lohmann et al., 2021). In particular, the rate of melting of the ice sheet on Greenland
was suggested to depend on the magnitude of the surface warming above its tipping point (Robinson et al., 2012). Such a rate–
induced cascade induced by crossing critical rates of environmental change complements the commonly suspected tipping
cascades involving bifurcation–induced tipping (Dekker et al., 2018; Klose et al., 2021; Wunderling et al., 2021).

## 3 Conceptual models describing individual tipping dynamics

In the following, we introduce conceptual process–based models representing the dynamics of the individual tipping elements.
The one–dimensional ice sheet model depicting the potential tipping behaviour of the Greenland Ice Sheet and the box model
capturing the AMOC thresholds are outlined in Sect. 3.1 and Sect. 3.2, respectively.

### 3.1 Greenland Ice Sheet evolution with a one–dimensional ice sheet model including melt–elevation feedback

To describe the behaviour of the GIS, we use a well–established flowline model in the $x$–$z$–plane, where the ice sheet rests
on a flat, rigid bed. Basal melting is neglected and the ice softness is assumed to be constant, i.e., it does not depend on the
temperature. The evolution of the ice thickness $h$ based on the shallow–ice approximation (Hutter, 1983) can then be described
by the following governing equation (Greve and Blatter, 2009):

$$\frac{\partial h}{\partial t} = -\frac{\partial}{\partial x}F + a_s \tag{1}$$

$$F = -\frac{2A(\rho g)^n}{n+2}\left|\frac{\partial h}{\partial x}\right|^{(n-1)}\frac{\partial h}{\partial x}h^{(n+2)} \tag{2}$$

with the surface mass balance $a_s$, ice softness $A$, Glen's flow law exponent $n$, the ice density $\rho$ and the gravitational acceler-
ation $g$. Changes in ice thickness $h$ depend on the divergence of the ice flux (first term on the right hand side of Eq. (1)) and
the mass balance at the surface $a_s$ (second term on right hand side of Eq. (1)). We assume a horizontal ice–sheet extent of $2L$
from $x = -L$ and $x = L$ being symmetric around the ice dome with zero ice thickness at the boundary (Jouvet et al., 2011, as-
sociated with a continent bounded by the ocean in Oerlemans (1981)). If not stated otherwise, the parameter values in Table S1
are used (representing conditions similar to present-day Greenland). The ice thickness equation Eq. (1)–(2) is combined with
a simple parameterization of the melt–elevation feedback (Zeitz et al., 2022) following Levermann and Winkelmann (2016) to
capture the non–linear dynamics and tipping behaviour of the GIS (Robinson et al., 2012). That is, a lowering of the ice sheet





surface enhances surface melt as the ice sheet surface is exposed to warmer air temperatures according to the atmospheric lapse rate $\Gamma$. Thereby, the surface mass balance $a_s$ is reduced and further ice loss is promoted. In particular, it is assumed that the surface mass balance $a_s$ depends linearly on the ice thickness $h$ (here equivalent to the ice sheet surface elevation) such that a changing ice thickness alters the surface mass balance as follows:

$$a_s = \tilde{a_0} + \gamma \Gamma h, \tag{3}$$

with the atmospheric lapse rate $\Gamma > 0$ and the surface melt sensitivity $\gamma$ describing the variation in surface melt with temperature changes (Levermann and Winkelmann, 2016). Based on the thickness $h(x,t)$ of the ice sheet with a horizontal extent 2L (Fig. 1(a)), the ice volume is approximated using a constant ice sheet length $w = 1000$ km (Fig. 1(a)). The value of the ice sheet length is chosen such that the present–day GIS ice volume (Morlighem et al., 2017) is approximately obtained for the initial ice sheet configuration at the start of our experiments. Note that the ice sheet length $w$ is kept constant irrespective of a

possible change of the GIS ice thickness $h(x,t)$.

The ice thickness evolution equation Eq. (1)–(2) together with the melt–elevation feedback Eq. (3) has been shown to generally capture the hysteresis behaviour of the GIS (Levermann and Winkelmann, 2016): For $0 = a_{0_{gc}} < \tilde{a_0}$ a stable ice sheet is built up, where $a_{0_{gc}}$ denotes the glaciation threshold. Two configurations of the ice sheet exist for $a_{0_{dgc}} < \tilde{a_0} < a_{0_{gc}}$, where the ice sheet will either evolve into a stable state with the ice volume close to present-day, or an ice–free state is obtained depending on the initial conditions. Crossing the deglaciation threshold $a_{0_{dgc}} > \tilde{a_0}$ leads to a complete disintegration of the ice

sheet. Note that the ice–free state is obtained by enforcing a non–negative ice thickness (Hindmarsh, 2001; Van den Berg et al., 2006). Obtaining a small remaining ice cap under warming as suggested by fully–dynamic ice sheet models (e.g., Robinson et al., 2012) requires to include additional processes beyond those considered here.

### 3.2  AMOC evolution using a box model of the global ocean

The dynamics of the AMOC is described by a global ocean box model (Wood et al., 2019; Alkhayuon et al., 2019), which consists of five boxes: the North Atlantic ($N$), the Tropical Atlantic ($T$) and the Indo–Pacific ($IP$) box connected via the Southern Ocean ($S$) box and a box corresponding to the bottom waters ($B$). Following Wood et al. (2019), it is assumed that the temperature $T_N$ of the North Atlantic box is linearly dependent on the AMOC strength $q$

$$T_N = \mu q + T_0 \tag{4}$$

with the North Pacific temperature $T_0$ and the constant $\mu$, while the temperatures of the other boxes are fixed. The AMOC strength $q$ is determined by the density difference between the North Atlantic and the Southern Ocean box

$$q = \lambda [\alpha(T_S - T_N) + \beta(S_N - S_S)] = \frac{\lambda[\alpha(T_S - T_0) + \beta(S_N - S_S)]}{1 + \lambda\alpha\mu} \tag{5}$$

where $\lambda$ is a hydraulic constant and $\alpha$ and $\beta$ are the thermal and haline coefficients, respectively (Wood et al., 2019).





By salt conservation, the salinities $S_i$ with $i \in \{N, T, S, IP, B\}$ for $q > 0$ are described by

$$V_N \frac{dS_N}{dt} = q(S_T - S_N) + K_N(S_T - S_N) - F_N S_0 \tag{6}$$

$$V_T \frac{dS_T}{dt} = q\left[\kappa S_S + (1-\kappa)S_{IP} - S_T\right] + K_S(S_S - S_T) + K_N(S_N - S_T) - F_T S_0 \tag{7}$$

$$V_S \frac{dS_S}{dt} = \kappa q(S_B - S_S) + K_{IP}(S_{IP} - S_S) + K_S(S_T - S_S) + \eta(S_B - S_S) - F_S S_0 \tag{8}$$

$$V_{IP} \frac{dS_{IP}}{dt} = (1-\kappa)q(S_B - S_{IP}) + K_{IP}(S_S - S_{IP}) - F_{IP} S_0 \tag{9}$$

and analogously for $q < 0$ with the box volumes $V_i$, the surface freshwater fluxes $F_i$ and the gyre coefficients $K_i$ as coefficients

of a diffusive flux representing a wind-driven salinity transport between the boxes where $i \in \{N, T, S, IP, B\}$. The parameter $\eta$

describes the mixing between the Southern Ocean and the bottom water box. $\kappa$ gives the proportion of the cold water path as

the AMOC flow returning via the South Pacific and the Drake Passage (Wood et al., 2019). The salinity $S_B$ in the bottom water

box is determined by assuming a constant total salt content ($C = $ const., determined by the initial conditions for the salinities

following Alkhayuon et al. (2019), compare Table S2)

$$C = V_N S_N + V_T S_T + V_S S_S + V_{IP} S_{IP} + V_B S_B \tag{10}$$

given that the surface freshwater fluxes satisfy $F_N + F_T + F_S + F_{IP} = 0$. A hosing $H$ resulting in the surface freshwater fluxes

of the form

$$F_i = F_{i_0} + A_i H \tag{11}$$

is applied where $i \in \{N, T, S, IP\}$ following Wood et al. (2019) with balanced surface freshwater fluxes as shown in Table S3.

The hosing pattern corresponds to an additional freshwater input into parts of the North Atlantic and Tropical Atlantic box (i.e.

the North Atlantic over 20–50° N) and a freshwater removal elsewhere. If not stated otherwise, the parameters displayed in

Table S1 are used. Note that freshwater fluxes are introduced as virtual salinity fluxes as in previous ocean box models, e.g.,

Rahmstorf (1996); Lucarini and Stone (2005) and likewise in some GCMs, e.g., Swingedouw et al. (2013); Yin et al. (2010);

Rahmstorf (1996) using a reference salinity. Thus, their effect on the mass balance is neglected keeping the ocean volume

constant.

## 4    Results

### 4.1    Modelling interactions of GIS and AMOC via freshwater fluxes and temperature

GIS and AMOC interact via freshwater fluxes into the North Atlantic originating from a melting GIS on the one hand, and via

a relative cooling around Greenland with a slowdown of the AMOC on the other. These suggested interactions are included in

our study by the coupling of the above described models as follows:

The relative cooling in the North Atlantic with a weakening of the AMOC (Vellinga and Wood, 2002, 2008; Jackson et al.,

2015) is assumed to imprint on the atmosphere and is related to the surface mass balance of the GIS via a constant factor $d_{oa}$





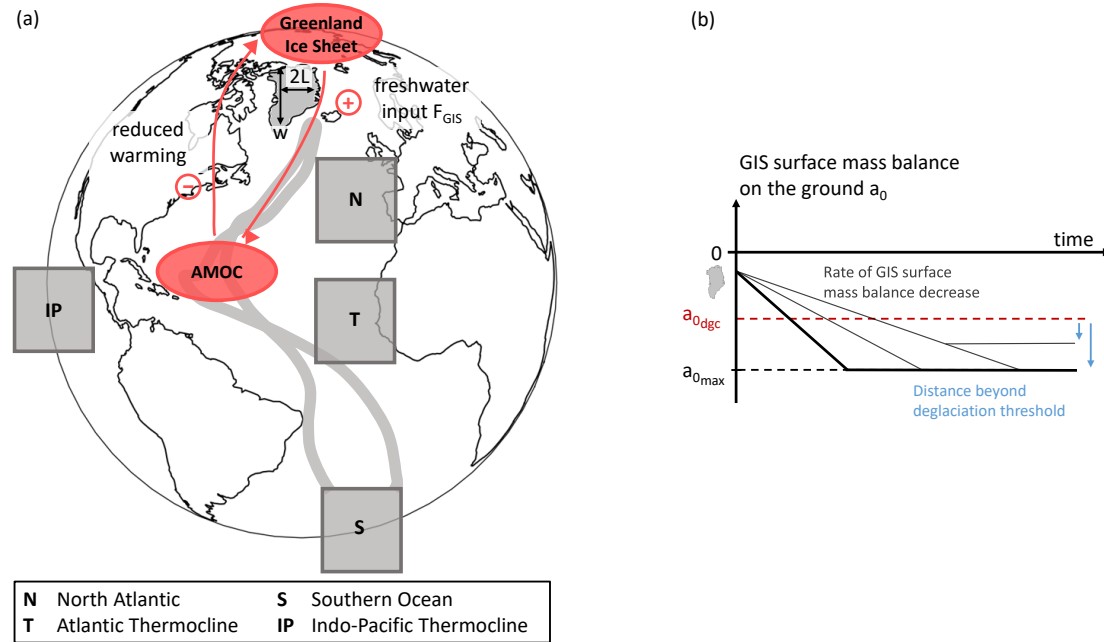

**Figure 1. Interactions between Greenland Ice Sheet (GIS) and Atlantic Meridional Overturning Circulation (AMOC).** (a): The model presented here investigates the positive–negative feedback loop between the two tipping elements via freshwater fluxes from Greenland ice loss and temperature changes due to changes in the overturning circulation. The dynamics of the Greenland Ice Sheet is modelled by a simplified approach including the melt–elevation feedback (see Eq. (1)-(3)). The ice sheet extent is captured by its horizontal width $2L$ and a constant length $w$, as indicated in the figure. The AMOC is represented by a box model (see Eq. (4)–(11)). (b): The GIS surface mass balance on the ground decreases linearly in time in our experiments across the deglaciation threshold $a_{0_{dgc}}$ with a ramping rate $r_{a_0}$ towards a final value $a_{0_{max}}$. Both the ramping rate $r_{a_0}$ and the final value $a_{0_{max}}$ are varied across the experiments presented here, as indicated by the distinct lines in (b).

and the ice melting sensitivity $\gamma$. The surface mass balance of the GIS $\tilde{a}_0$ in Eq. (3) is then replaced by

$$\tilde{a}_0 = a_0 + \gamma d_{oa}(T_{N_{H_{ref}}} - T_N) \tag{12}$$

where $T_{N_{H_{ref}}}$ is a reference temperature in the North Atlantic box given with respect to a reference hosing $H_{ref}$. We will refer to $a_0$ as the surface mass balance on the ground. In the following, $H_{ref} = 0$ Sv is chosen corresponding to the quasi–equilibrated AMOC under preindustrial atmospheric $CO_2$ concentration conditions. For a declining overturning strength $q$ of the AMOC with $H > H_{ref}$, the temperature $T_N$ in the North Atlantic box declines as well according to Eq. (4). For $d_{oa} = 0$, we recover a

unidirectional coupling with an independent ice sheet, whose evolution influences the AMOC via freshwater fluxes specified in the following.





The freshwater flux into the ocean along Greenland's coast resulting from the mass loss of the GIS (Bamber et al., 2012, 2018; Trusel et al., 2018) is added as $F_{\mathrm{GIS}}$ to the surface freshwater flux hosing of the North Atlantic box as:

$$F_N = F_{N_0} + A_N H + F_{\mathrm{GIS}} \tag{13}$$

The GIS freshwater flux $F_{\mathrm{GIS}}$ is determined by integrating the thickness change of the ice sheet over its spatial horizontal extent and approximated into a volume loss by the constant ice–sheet length $w$ (Sect. 3.1). It eventually acts as a virtual salinity flux, while assuming a constant ocean volume (compare Section 3.2). The freshwater flux $F_{\mathrm{GIS}}$ from the GIS is set to zero ($F_{\mathrm{GIS}} = 0$ Sv) if the GIS resides in a steady–state configuration (or grows). Hence, the freshwater flux $F_{\mathrm{GIS}}$ is non–zero ($F_{\mathrm{GIS}} \neq 0$ Sv) only during a height (or volume) loss of the GIS over time corresponding to the ice sheet decline.

The freshwater flux $F_{\mathrm{GIS}}$ supplements the hosing $H$ and additionally controls the long–term stability of the AMOC. It has an additive effect on the total freshwater flux into the ocean, which enhances the already existing hosing $H$ and thus may take the AMOC to its 'off'–state if reaching a critical value throughout the GIS decline. As exemplarily indicated in Fig. 2(a) (black and grey lines), for a fixed hosing there exists a critical threshold $F_{\mathrm{GIS_{Hopf}}}(H = \text{const.})$ on varying the freshwater flux $F_{\mathrm{GIS}}$ beyond which the 'on'–state of the AMOC is not stable anymore. In particular, the upper stable branch loses stability via a subcritical

Hopf bifurcation at $F_{\mathrm{GIS_{Hopf}}}$ (indicated by green point in Fig. 2(a)). The upper branch disappears when it meets the unstable middle branch at a turning point of the bifurcation curve. Note that the Hopf bifurcation $F_{\mathrm{GIS_{Hopf}}}$ and the turning point are very close to each other and therefore cannot be clearly distinguished in Fig. 2(a). Figure 2(b) illustrates how the GIS freshwater flux threshold $F_{\mathrm{GIS_{Hopf}}}$ changes depending on the hosing $H$. With increasing hosing $H$ and thus by getting closer to the hosing threshold $H_{\mathrm{Hopf}}$ (Alkhayuon et al., 2019), the threshold $F_{\mathrm{GIS_{Hopf}}}$ is shifted to smaller values. Note that, while the GIS freshwater

flux $F_{\mathrm{GIS}}$ has been discussed in the style of an external control parameter here, it is eventually attributed to a time–dependent decline of the GIS and in fact turns into a state variable in transient experiments.

We explore the dynamics and possible tipping outcomes of the interacting GIS and AMOC, which are represented by the model introduced in Sect. 3 and coupled via freshwater fluxes and temperature as outlined above, in response to a changing surface mass balance on the ground $a_0$, as observed over the past decades and projected with progressing global warming

(Shepherd et al., 2020; van den Broeke et al., 2017; Fettweis et al., 2013). More specifically, the surface mass balance on the ground $a_0$ is decreased linearly with a ramping rate $r_{a_0}$ towards or across the deglaciation threshold $a_{0_{\mathrm{dgc}}}$ beyond which a stable ice sheet cannot be sustained. It is kept constant after a final value $a_{0_{\max}}$ is reached (Fig. 1(b)). The AMOC hosing $H < H_{\mathrm{Hopf}}$ is fixed. It is assumed that the ice sheet initially resides in a state with an intact ice sheet.

## 4.2    Tipping cascades between GIS and AMOC

In a first step, we study the AMOC response to a disintegration of the GIS under idealized forcing scenarios (as described above and indicated in Fig. 1(b)), complementing previous freshwater hosing experiments by choosing a coupling strength $d_{oa} = 0$. Different types of cascading tipping can be identified (Sect. 4.2.1). The occurrence of these qualitatively different tipping pathways is quantified in the space of parameters which determine the evolution of the environmental drivers for GIS and AMOC (Sect. 4.2.2).





### 4.2.1 Types of tipping cascades

By decreasing the surface mass balance on the ground associated with progressing warming as qualitatively displayed in Fig. 1(b), the GIS is forced across its deglaciation threshold and eventually melts down completely when neglecting the negative temperature feedback. The AMOC hosing $H$ is kept constant. For all experiments, it is assumed that the GIS initially resides in a steady state with an intact ice sheet for a surface mass balance on the ground $a_0 = -0.3 \, \mathrm{m \, a^{-1}}$ and the AMOC is initially in its 'on'–state corresponding to the fixed hosing $H = \mathrm{const}$. The approximated volume loss resulting from the forced deglaciation of Greenland is introduced as a time–varying GIS freshwater flux $F_{\mathrm{GIS}}$ into the North Atlantic. Thus, with the time–dependent GIS freshwater flux, which increases and subsequently decreases during the decline of the GIS, the AMOC moves in the $F_{\mathrm{GIS}}$– direction towards higher values of the freshwater flux $F_{\mathrm{GIS}}$, potentially overshooting its threshold (Ritchie et al., 2021), but eventually returns to $F_{\mathrm{GIS}} = 0 \, \mathrm{Sv}$ under a constant hosing (Fig. 2(a), with AMOC trajectory approximately following black and grey lines). For exemplary melting patterns of the GIS and positions of the AMOC relative to its hosing threshold we can identify different types of cascading tipping of the GIS and the AMOC. The identified patterns of cascading tipping are qualitatively comparable to AMOC responses to an artificial freshwater flux as observed in previous hosing experiments using GCMs.

In particular, the AMOC may transition to its 'off'–state in response to the loss of the Greenland Ice Sheet and accompanied by a temporary overshoot of the GIS freshwater flux threshold in an *overshoot cascade* (Fig. 2(c)). The increasing GIS freshwater flux takes the AMOC out of the basin of attraction of the 'on'–state and the AMOC does not return after the decline of the GIS freshwater flux with the deglaciation of Greenland. The surface mass balance is decreased strongly beyond the deglaciation threshold to $a_{0_{\mathrm{max}}} = -3.0 \, \mathrm{m \, a^{-1}}$ within 3000 years, which results in a complete deglaciation of Greenland in this time period. The resulting GIS freshwater flux is sufficiently slow such that the AMOC closely follows its 'on'–state. Note that the AMOC is already shifted towards its hosing threshold along the upper stable branch with a hosing $H = 0.16 \, \mathrm{Sv}$. Hence, the overshoot cascade does not necessarily contradict the AMOC weakening (without tipping), which is commonly observed in hosing experiments (Mikolajewicz et al. (2007), compare Sect. 4.2.2 for further discussion).

Under a more extreme collapse of the GIS within about 1000 years (for a faster and stronger decrease of the surface mass balance), the AMOC may undergo a critical transition to its 'off'–state without a crossing of the GIS freshwater flux threshold in a *rate–induced cascade* (Fig. 2(d)) as recently described for the AMOC due to an abrupt decline in sea–ice cover (Lohmann et al., 2021). With the relatively fast deglaciation of Greenland, the AMOC looses track of the stable 'on'–state and crosses the moving basin boundary. Rate–induced transitions of the AMOC have already been explored by Stocker and Schmittner (1997) for varying $CO_2$ emission rates. More recently, Lohmann and Ditlevsen (2021) confirmed the suggested sensitivity of the AMOC to the rate of change of freshwater fluxes by demonstrating rate–induced tipping in a complex ocean model. Here it is assumed that both the ice sheet on Greenland and the AMOC are initially in equilibrium. However, small disturbances e.g. in initial box salinities are always present in the real world. Initial conditions may additionally be important for the response of the AMOC to a GIS decline as studied e.g. as scenario–dependent basins of attraction (Kaszás et al., 2019).





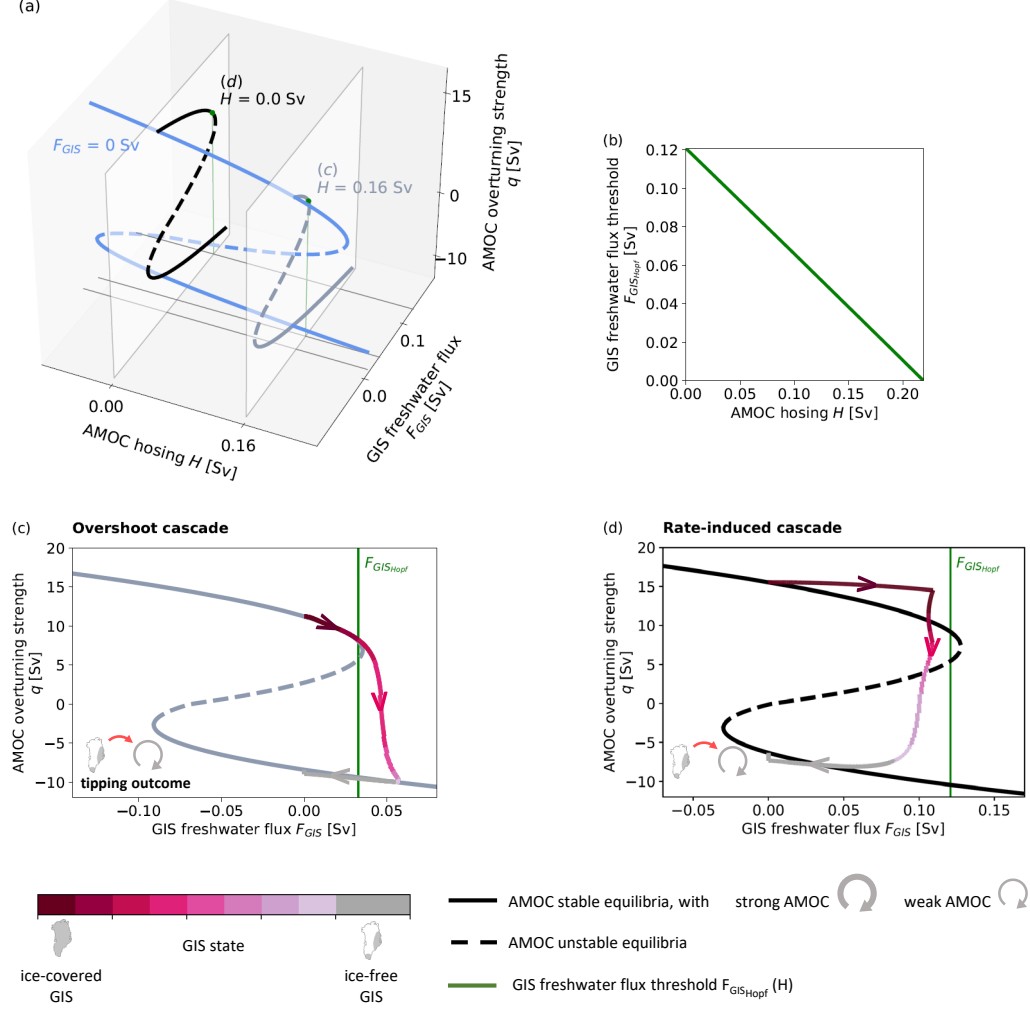

**Figure 2. Cascading tipping of the Greenland Ice Sheet and the Atlantic Meridional Overturning Circulation for unidirectional coupling.** (a): Long-term behaviour of the AMOC overturning strength $q$ as a function of the hosing $H$ and the GIS freshwater flux $F_{\text{GIS}}$. The uncoupled case with zero freshwater flux $F_{\text{GIS}} = 0$ Sv is indicated in blue; two cases under varying GIS freshwater flux with constant hosing $H = 0$ Sv and $H = 0.16$ Sv are shown in black and light grey, respectively. Stable fixed points are given by the solid lines, while unstable fixed points are given by the dashed lines. The critical GIS freshwater flux threshold $F_{\text{GIS}_{\text{Hopf}}}$ for AMOC hosing $H = 0$ Sv and $H = 0.16$ Sv is indicated in green. (b): GIS freshwater flux threshold $F_{\text{GIS}_{\text{Hopf}}}$ depending on the AMOC hosing $H$. (c) and (d): Response of the AMOC (pink to grey colouring indicating the respective state of the GIS at that point in time) in terms of the overturning strength $q$ to the deglaciation of Greenland and the resulting freshwater flux $F_{\text{GIS}}$ for a constant hosing $H$. The negative feedback via a relative cooling around Greenland is neglected with a coupling strength $d_{\text{oa}} = 0$. (c): *Overshoot cascade* for $H = 0.16$ Sv, $r_{a_0} = $ -0.001 m a$^{-2}$ and $a_{0_{\text{max}}} = $ -3.0 m a$^{-1}$, leading to tipping of the AMOC in response to a deglaciation of Greenland. (d): *Rate–induced cascade* for $H = 0$ Sv, $r_{a_0} = $ -0.1 m a$^{-2}$ and $a_{0_{\text{max}}} = $ -3.55 m a$^{-1}$, where the AMOC tips in response to the rapid ice loss from Greenland albeit not having crossed its own respective tipping point yet.



### 4.2.2 Emergent dynamic regimes

We find qualitatively different cascading dynamics of an AMOC transition in response to a deglaciation of Greenland in our
model as an *overshoot cascade* and a *rate–induced cascade*. The conceptual nature of the model allows to study these cascading
dynamics with respect to the GIS melting time scales as well as the AMOC position relative to its hosing threshold. By varying
related parameters in a next step, we thus systematically explore and quantify the occurrence of the tipping outcomes and the
regimes of the Greenland Ice Sheet and AMOC characterized by qualitatively different tipping dynamics. Thereby, we are able
to qualitatively identify safe and dangerous pathways (Armstrong McKay et al., 2022) for the evolution of the tipping element
drivers.

The deglaciation of Greenland in response to an idealized linear decrease of the surface mass balance on the ground
(Fig. 1(b)) is determined by how fast (rate of change of the surface mass balance of the ground $r_{a_0}$) and how far (final value
beyond the deglaciation threshold $a_{0_{\max}}$) the Greenland Ice Sheet is driven across its tipping point. Figure 4 shows the overall
tipping outcome (indicated by the colouring) depending on the timescale of GIS decline (by varying the rate of change of the
surface mass balance $r_{a_0}$ along the outer vertical axis and the final value of the surface mass balance $a_{0_{\max}}$ along the outer
horizontal axis). In addition, the distance of the AMOC to its hosing threshold is taken into account (by varying the constant
hosing $H$ from $H = 0\,\mathrm{Sv}$ to close to the hosing threshold $H_{\mathrm{Hopf}}$ along the vertical axis of the respective bar). The hosing value
above which additional freshwater from Greenland gives rives to the stability loss of the AMOC 'on'–state (compare Fig. 2(a)
and (b)) is denoted by the green line in Fig. 4. We are thus able to detect a rate–induced collapse of the AMOC, which occurs
before the strong AMOC state loses stability and hence without crossing critical magnitudes of freshwater flux.

For slowly driving the Greenland Ice Sheet slightly across its deglaciation threshold (lower left corner in Fig. 3), the oc-
curence of the *overshoot cascade* with an overshoot of the GIS freshwater flux threshold (compare Fig. 2(c)) is limited to
relatively high hosing values sufficiently close to the AMOC hosing threshold (solid grey area above green line). For rela-
tively lower hosing values and thus for the AMOC residing in greater distance to its hosing threshold, the AMOC temporarily
weakens with freshwater input from Greenland but eventually remains in its 'on'–state (as commonly observed in hosing ex-
periments) in response to a slow GIS deglaciation. The GIS freshwater flux threshold is not crossed (dashed grey area below
green line). Thus, for an overshoot cascade to occur with a slow ice sheet decline a high hosing determining the fixed surface
freshwater flux hosing pattern is necessary in addition to the freshwater from the ice sheet on Greenland. In other words, the
AMOC is to be shifted closer to its hosing tipping point for a propagation of tipping.

The relative size of the region in the parameter space which gives rise to an overshoot cascade changes by varying the
melting patterns of the GIS. More specifically, with a faster decrease of the surface mass balance and an increasing distance
beyond the deglaciation threshold of Greenland (going from lower left corner to center of Fig. 3, resulting in a more rapid ice
sheet collapse) the overshoot cascade is found already for lower values of the hosing $H$ (solid grey area above green line).
Hence, an AMOC collapse due to overshooting the respective tipping point with a GIS deglaciation may already occur for
larger distances of the AMOC to its hosing tipping point.





Finally, a more rapid ice sheet decline with a fast onset of GIS melting and a sufficiently long period of sustained, high freshwater input from Greenland allows for a *rate–induced cascade* to emerge (compare Fig. 2(d)). The AMOC collapses due to the rapid ice loss from Greenland without having crossed its respective tipping point $F_{\mathrm{GIS_{Hopf}}}$ (going from center to upper right corner of Fig. 3, solid grey area below green line).

The ocean box model (Wood et al., 2019) may additionally allow for avoiding an AMOC collapse despite overshooting the respective tipping point. Such a safe overshoot requires a fast onset of GIS melting followed by a fast enough decrease of the freshwater flux (Alkhayuon et al., 2019; Wunderling et al., 2023). Starting from a Greenland Ice Sheet which approximately resembles present–day conditions, safe overshoots of the AMOC tipping point are not found in our model for the range of melt time scales considered here.

### 4.3   Limited potential for stabilization with additional negative feedback

Finally, we explore the suggested stabilization effect of the additional negative feedback from a relative cooling around Greenland with a weakened AMOC (Gaucherel and Moron, 2017) for the overall system behaviour. The negative feedback via temperature was omitted when identifying the types of cascading tipping and determining their occurrence in the parameter space characterizing the evolution of environmental drivers for the Greenland Ice Sheet and the Atlantic Meridional Overturn-

ing Circulation (Sect. 4.2).

Considering this additional negative feedback, the intrinsic tipping point of the Greenland Ice Sheet (that is, the critical threshold of the Greenland Ice Sheet without any coupling, compare Klose et al., 2020) is replaced by two distinct effective deglaciation thresholds $a_{0_{\mathrm{dgc}}}^{(1)}$ and $a_{0_{\mathrm{dgc}}}^{(2)}$ of the GIS. More specifically, interactions shift the critical threshold of the system beyond which tipping is expected to lower or higher values compared to the intrinsic tipping point depending on the direction

of coupling and the state of the influencing tipping element, giving rise to effective tipping point(s) (Klose et al., 2020). Here, the effective deglaciation thresholds arise when taking into account the stabilizing effect of an AMOC weakening and may be crossed with a decreasing surface mass balance, that is projected for a warming climate (Fettweis et al., 2013). For $a_0 < a_{0_{\mathrm{dgc}}}^{(1)}$ a complete melting of the GIS can be observed given that the AMOC resides and remains in its 'on'–state. Given that the AMOC resides in its 'off'–state, the GIS melts down completely for $a_0 < a_{0_{\mathrm{dgc}}}^{(2)}$. These distinct tipping thresholds suggest a limited

decrease of the surface mass balance of the ground to $a_{0_{\mathrm{dgc}}}^{(1)} >> a_{0_{\mathrm{max}}} > a_{0_{\mathrm{dgc}}}^{(2)}$ as well as a strong decrease of the surface mass balance on the ground $a_{0_{\mathrm{max}}} << a_{0_{\mathrm{dgc}}}^{(2)}$ beyond the effective deglaciation threshold $a_{0_{\mathrm{dgc}}}^{(2)}$ as different scenarios.

Decreasing the surface mass balance emulating a warming climate beyond its effective threshold $a_{0_{\mathrm{dgc}}}^{(2)}$ (corresponding to a strong surface mass balance decrease) does not allow for a GIS stabilization (Fig. 4(a) and (b)). Instead, for an AMOC residing sufficiently close to its hosing threshold, a GIS deglaciation and tipping of the AMOC to the 'off'-state is observed (trajectory

for $H = 0.205$ Sv in Fig. 4(a) and corresponding grey area in Fig. 4(c)). Given a lower freshwater hosing $H$, the AMOC remains in its 'on'–state with the deglaciation of the GIS (trajectory for $H = 0.16$ Sv in Fig. 4(b) and corresponding dashed grey area in Fig. 4(c)). Hence, for a strong surface mass balance decrease the tipping outcomes in terms of the final GIS and AMOC states when neglecting the negative feedback via the temperature are qualitatively resembled (Fig. 4(c)).





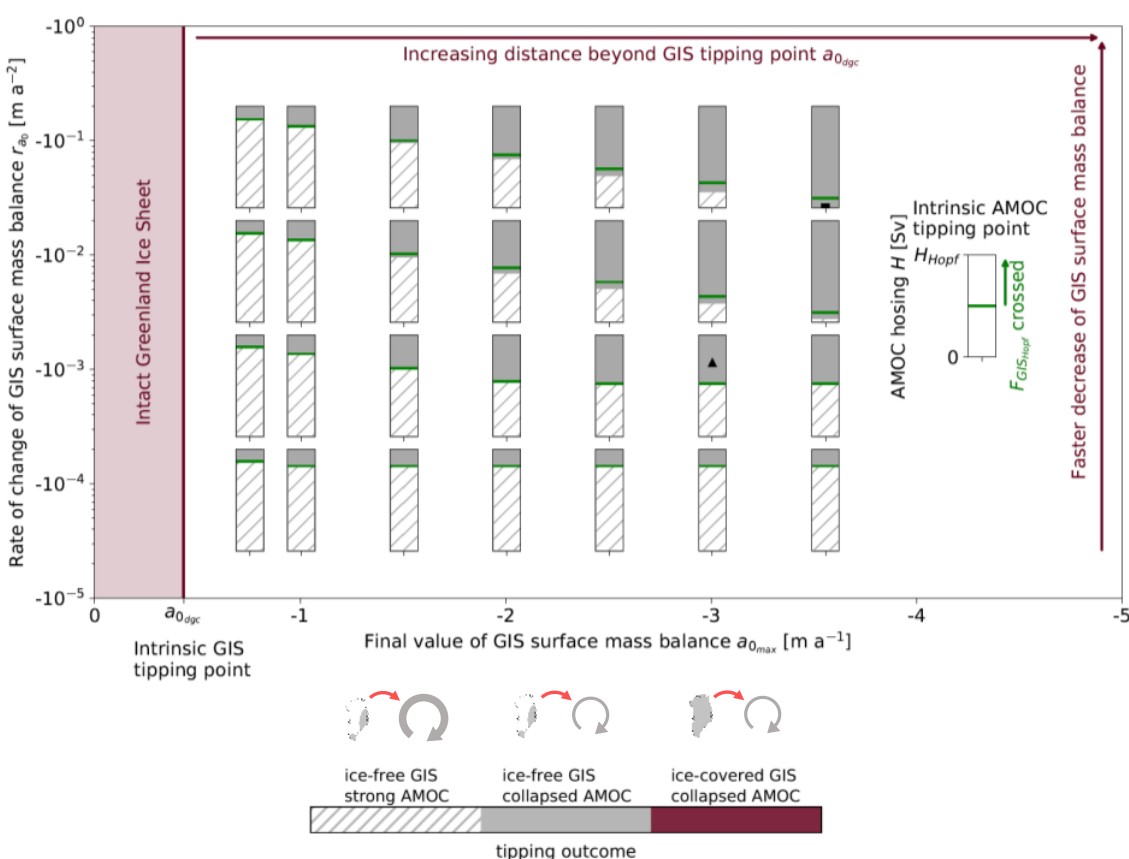

**Figure 3. Emergent dynamic regimes of the Greenland Ice Sheet and the Atlantic Meridional Overturning Circulation for unidirectional coupling.** Tipping outcomes in response to a GIS decline by linearly decreasing its surface mass balance on the ground (associated with progressing warming) with a ramping rate $r_{a_0}$ (varied along outer vertical axis) to a final value $a_{0_{max}}$ (varied along the outer horizontal axis) beyond the GIS deglaciation threshold. The AMOC hosing (vertical axis of bars) is kept constant between $H = 0$ Sv and the AMOC hosing threshold $H_{Hopf}$. The respective tipping outcome is indicated by the colouring (grey: GIS deglaciation, pink: no GIS deglaciation; stripes additionally indicate the AMOC in its 'on'–state). The hosing above which the GIS freshwater flux threshold $F_{GIS_{Hopf}}$ is crossed temporarily by the freshwater flux arising from the GIS decline is indicated by the green line within in each bar. The black diamond and the black rectangle indicate the combination of tipping element drivers for the overshoot cascade and rate–induced cascade, respectively, as displayed in Fig. 2(c) and (d).





A limited decrease of the surface mass balance may allow for a GIS stabilization by the negative temperature feedback.
As exemplarily shown for a constant AMOC hosing $H = 0.205$ Sv in Fig. 4(d), the AMOC may leave its 'on'–state and approach its 'off'–state with an initial melting event of the GIS. With the AMOC tipping a relative cooling of the North Atlantic box follows from the assumed linear dependence of the North Atlantic box temperature from the AMOC overturning strength. Eventually, the GIS does not continue melting after the initial melting event (compare colour coding in Fig. 4(d)). The deglaciation of Greenland is avoided and the ice sheet is stabilized (for at least the time period covered by the simulations)
by the tipping AMOC in response to a pronounced initial melting. However, the AMOC is required to reside close to its hosing threshold for the GIS stabilization to unfold and additionally to undergo a critical transition itself as indicated by the stabilization corridor (Fig. 4(f), pink corridor). For pathways of a (limited) surface mass balance decrease outside of this stabilization corridor, the GIS melts down completely while the AMOC remains in its 'on'–state (trajectory for $H = 0.16$ Sv in Fig. 4(e) and corresponding corridor in Fig. 4(f), dashed grey area).

**5  Discussion and Conclusion**

In summary, qualitatively distinct cascading dynamics may arise from the interaction of the Greenland Ice Sheet and the Atlantic Meridional Overturning Circulation in a positive–negative feedback loop as suggested by a process–based conceptual model. The model captures the main positive feedback mechanisms for the potential tipping behaviour of both tipping elements as well as their interaction via ice loss from Greenland introduced into the North Atlantic and a net–cooling around Greenland
with an AMOC weakening. Accompanied by a temporary overshoot of its critical threshold by the freshwater flux from a deglaciation of the Greenland Ice Sheet, the AMOC may undergo a critical transition in our model in an *overshoot cascade* on the one hand. By contrast, tipping of the AMOC may occur without the exceedance of the GIS freshwater flux threshold in a *rate–induced cascade* given a fast onset of GIS decline. Finally, an unfolding of the negative feedback via a relative cooling around Greenland and a stabilization of the ice sheet is conditional on an AMOC collapse in our model. Our results stress
that the interplay of applied external and corresponding internal forcing time scales relative to the response time scales of the tipping elements is of importance for interacting tipping elements of the climate system as theses time scales may eventually determine the tipping dynamics.

Accordingly, the occurrence of qualitatively distinct tipping dynamics and outcomes vary with the ice sheet melting time scales. This implies that safe pathways for the evolution of tipping element drivers preventing cascading tipping and their
boundary to dangerous pathways involving cascades are controlled by rates of changes of the responsible control parameters in addition to their magnitude. Hence, our model qualitatively suggests that it is not only necessary to stay below critical thresholds in terms of the magnitude of some environmental condition (Schellnhuber et al., 2016) as intended by the Paris Agreement (UNFCCC, 2015) to hinder tipping cascades. In addition, it is required to respect safe rates of environmental change to mitigate domino effects as concluded previously for individual tipping elements (Ashwin et al., 2012; Luke and
Cox, 2011; Petschel-Held et al., 1999; Stocker and Schmittner, 1997; Wieczorek et al., 2011; Schoenmakers and Feudel, 2021)





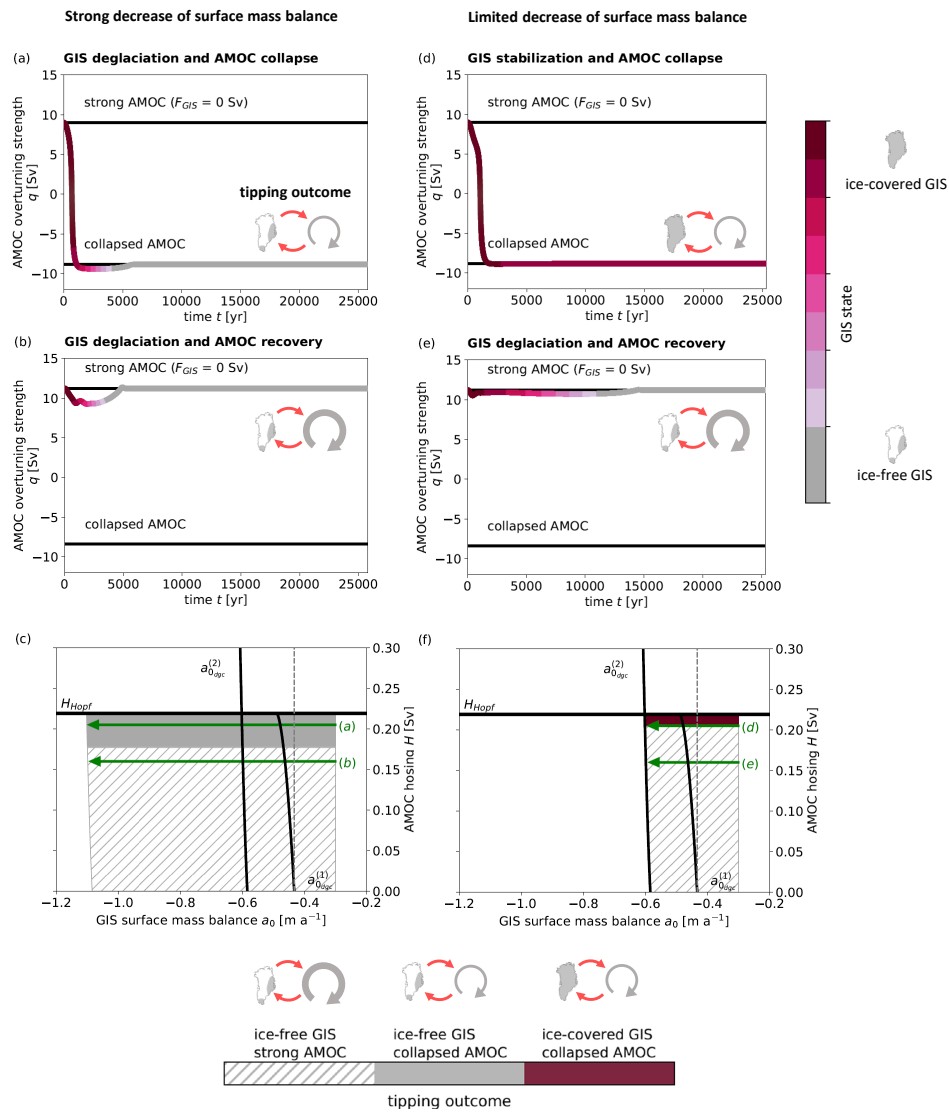

**Figure 4. Tipping dynamics for bidirectional coupling between Greenland and the AMOC.** Shown is the AMOC overturning strength, now also taking into account the negative feedback via relative cooling around Greenland with a coupling strength $d_{oa} = 2.857$ for a ramping rate $r_{a_0} = $ -0.001 m a$^{-2}$ with a strong decrease of the GIS surface mass balance (left column) and a limited decrease of the GIS surface mass balance (right column) under a constant hosing $H$. (a)–(b) & (c)–(d): Dynamics of the AMOC in terms of the overturning strength $q$ over time. In addition, the GIS state in terms of the percentage of the initial GIS ice volume is indicated by colouring declining from pink (100 %) to grey (0 %). The black lines indicate the 'on'– and the 'off'–state of the AMOC for the respective constant hosing without an additional freshwater input from Greenland ($F_{GIS} = 0$ Sv). (c) & (f): Tipping outcomes of GIS and AMOC for pathways of surface mass balance decrease with distinct constant hosing $H$ within the $(a_0, H)$–plane. The respective tipping outcome is indicated by the colouring (grey: GIS deglaciation, pink: no GIS deglaciation; stripes additionally indicate the AMOC in its 'on'–state). Solid black lines indicate the critical thresholds of the GIS and the AMOC. The intrinsic thresholds $a_{0_{dgc}}$, which arises by neglecting the coupling via the temperature with a coupling strength $d_{oa} = 0$, is indicated as grey dashed lines.





but still pending to be incorporated in management strategies to maintain the resilience of the Earth system (Rockström et al., 2009; Steffen et al., 2015; UNFCCC, 2015; Rockström et al., 2023).

The Greenland Ice Sheet is at risk of crossing its tipping point with >1.5°C global warming (Robinson et al., 2012; Armstrong McKay et al., 2022). At present, the ice sheet's mass loss is accelerating (Shepherd et al., 2020) and its western part already shows early warning signs of approaching a critical transition (Boers and Rypdal, 2021). While the crossing of the critical temperature threshold itself does not imply a fast collapse, the time needed to melt the ice sheet on Greenland decreases with a higher temperature level above its tipping point (as qualitatively obtained with our model as well as quantified using a three–dimensional polythermal ice sheet model by Robinson et al., 2012). As a consequence, the future level of warming (even if having transgressed the threshold) controls the rates of mass loss from Greenland and is thereby, among others, decisive for its impacts on cascading tipping of the AMOC.

In addition, the fate of the AMOC in response to freshwater input from the Greenland Ice Sheet is strongly dependent on the AMOC position relative to the hosing threshold in our model. Given that the AMOC remains relatively far from its hosing threshold, it may remain in its currently attained strong state. However, shifting the AMOC towards its hosing threshold (e.g., with increasing precipitation in the North Atlantic) can bring it into a region where freshwater from a GIS decline may induce a collapse (either by overshooting the respective tipping point or with a fast onset of GIS melting). This suggests that AMOC weakening in hosing experiments and the inferred risk of an AMOC collapse with ongoing global warming has to be evaluated from a dynamical systems point of view (compare Weijer et al., 2019) and with respect to the distance of the present–day AMOC to its tipping point (which is still relatively unknown). At the same time, the AMOC may already be shifted closer to its tipping point: A decline of 15 % of the overturning circulation since the mid–twentieth century is inferred from the sea–surface temperature fingerprint (Caesar et al., 2018) and common early warning indicators indicate an ongoing loss of stability (Boers, 2021).

Utilizing idealized (Dekker et al., 2018; Klose et al., 2020; Wunderling et al., 2021) or process–based while conceptual representations of climatic tipping elements (such as by Dekker et al., 2018; Lohmann et al., 2021, and as for the Greenland Ice Sheet and the AMOC here) allows to qualitatively understand possible cascading dynamics in the Earth's climate system arising from tipping element interactions on long time scales. At the same time, conclusions to be drawn are limited because of simplifications both in the representation of the individual tipping elements, e.g. by a one–dimensional ice sheet on a flat bed, and in their coupling, e.g. by the approximation of freshwater fluxes. Further extending the presented conceptual model capturing the interactions of the GIS and the AMOC by an evolution of ocean box temperatures or by adding climatic tipping elements and their respective interactions may enable a probabilistic assessment of the risk of cascading behaviour in the network of tipping elements under global warming taking into account uncertainties. For example, an additional freshwater flux into the Southern Ocean from a retreat of the West Antarctic Ice Sheet may prevent a collapse of the AMOC despite of a deglaciation of Greenland under certain conditions as suggested recently by a model of comparable complexity (Sinet et al., 2023). The stabilizing effect of a net–cooling around Greenland with an AMOC weakening is, however, not included in the conceptual model of Sinet et al. (2023). To the end, we are still lacking quantitative insights on (1) the position of climatic tipping elements under current climate conditions with respect to their tipping points, (2) the strength of their interactions and,



subsequently, (3) the role of tipping cascades in the future evolution of the Earth system, in particular under global warming. These may be obtained given an ongoing improvement of climate models e.g. by including ice sheet dynamics (De Rydt and Gudmundsson, 2016; Gierz et al., 2020; Kreuzer et al., 2021). Finally, linking modelling approaches to modern but also paleoclimate data (Thomas et al., 2020) may help to reduce uncertainties on the emergence of tipping cascades in the past and

in the future.

## 6   Code and data availability

Code and data used for producing the results in this study will be archived within the open access repository Zenodo upon publication of the manuscript.

*Author contributions.*   All authors conceived of and designed the study. A.K.K performed the analysis, led the writing of the manuscript and
created figures and tables. All authors provided feedback on the analysis and input to the manuscript.

*Competing interests.*   One of the (co-)authors is a member of the editorial board of Earth System Dynamics. The authors have no other competing interests to declare.

*Acknowledgements.*   This work has been performed in the context of the FutureLab on Earth Resilience in the Anthropocene at the Potsdam Institute for Climate Impact Research. The authors gratefully acknowledge the European Regional Development Fund (ERDF), the German
Federal Ministry of Education and Research (BMBF) and the Land Brandenburg for supporting this project by providing resources on the high-performance computer system at the Potsdam Institute for Climate Impact Research. A.K.K and R.W acknowledge support by the European Union's Horizon 2020 research and innovation programme under Grant Agreement No. 820575 (TiPACCs) and No. 869304 (PROTECT). J.F.D. is grateful for financial support by the Leibniz Association (project DominoES), the European Research Council (ERC) under the European Union's Horizon 2020 Research and Innovation Programme (ERC Grant Agreement No. 743080 ERA) and BMBF
within the framework 'PIK Change' (Grant No. 01LS2001A). U.F. acknowledges support by the European Union's Marie Sklodowska–Curie Innovation Training Network under grant agreement No. 956170 (Critical Earth). A.K.K thanks Maria Zeitz and Ronja Reese for helpful discussions.



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
