# Peer review of "Rate-induced tipping cascades arising from interactions between the Greenland Ice Sheet and the Atlantic Meridional Overturning Circulation"

_Earth System Dynamics, 2023_

## Referee Comment (RC3)

*Review of* Klose, Donges, Feudel, and Winkelmann*: "Rate–induced tipping cascades arising from interactions between the Greenland Ice Sheet and the Atlantic Meridional Overturning Circulation." (*Earth System Dynamics*, Paper:* esd-2023-20*)*

The manuscript of Klose and others presents the results of a conceptional model system where a global ocean interacts with the Greenland Ice Sheet (GrIS). They allow tipping cascades of the Atlantic Meridional Overturning Circulation (AMOC) stability by freshwater released into the North Atlantic. Besides a hosing flux, the meltwater contribution of disintegrating GrIS disturbs the stability of the AMOC and, via a potential feedback loop, controls Greenland's meltwater contribution. The authors access overshot and rate-induced tipping cascades in this highly idealized system.

The first conceptional model renders the global ocean by hydrographic properties in five boxes: Two dedicated Atlantic surface ocean regions (North Atlantic: N, Tropical Atlantic: T) beside surface boxes for the Southern Ocean (S) and the Indo-Pacific Ocean (IP). Those surface boxes communicate with a global bottom ocean box (B). In this model, the density difference between the Northern Atlantic and Southern Ocean drives the AMOC, which controls the North Atlantic temperature and determines the temporal salt flux between different boxes (Eq. 6 – 10).

The other conceptional model represents the Greenland Ice Sheet by a flowline model solving the shallow ice approximation. It has a (half) width of 1000 km, which equals approximately its actual latitudinal extent. The ice loss and gain are exclusively described by the surface mass balance (SMB), where the Lapse rate effect constitutes the melt elevation feedback.

The coupling between these two models is unidirectional or bidirectional. The unidirectional coupling considers only the meltwater flux of a shrinking GrIS into the Atlantic Ocean. This meltwater flux plus an additional freshwater hosing flux decreases the salinity in the Atlantic. In the bidirectional setup, the North Atlantic temperature feeds back on Greenland's surface mass balance.

The authors detect overshoot and rate-induced cascading tipping in their model system with a focus on the GrIS and the AMOC. These cascades are analyzed in the uni- and bidirectional setups where the bidirectional coupling considers the feedback loop between the North Atlantic temperature and the ice loss. This thermal coupling stabilizes the AMOC and the ice sheet because an enhanced meltwater flux reduces, via the density difference, the AMOC strength, lowering the North Atlantic temperatures, which, ultimately, damps additional melting.

This study is highly relevant since it addresses outstanding questions about the AMOC stability while considering the interaction between the AMOC and the Greenland Ice Sheet. At the same time, Greenland's meltwater release grows in a warming world. The authors highlight that it is not sufficient to consider a "fixed" threshold (overshoot tipping) beyond which the AMOC breaks down or GrIS disintegrates. They also underscore that changing rates could drive components in the coupled system beyond stable conditions (rate-induced tipping). In addition, a stabilizing feedback, such as reduced North Atlantic warming due to a weakened AMOC, may not be strong enough to offset the disintegration of the GrIS. Here, the reduced model complexity allows scanning the phase space for numerous tipping cascades. Although, these are not necessarily representative of the natural world.

In general, it was a pleasure to read the well-structured manuscript. The figures are of high quality, necessary, and informative.

**I recommend the publication of the manuscript after minor corrections.**

**General comments**

Although the manuscript is well organized and generally well written, section 4.2, including its subsections, could be better written. The language of section 4.2 is less clean than the remaining section. Therefore, I suggest revising this whole section.

Furthermore, the authors presented additional lengthy information in brackets, which may disturb the reader. I suggest the authors integrate this information into the general text or drop it if applicable.

In Figures three and four, small schematic icon-like figures of the remaining ice-thickness across Greenland seem to indicate remaining ice. If this is the case, please state more clearly where this pattern comes from because I do not see how the applied flowline ice sheet model can provide this pattern.

Occupationally, terms/variables are introduced, which are defined later in the text. In some cases, this needs to be clarified. For example, the text refers (Page 9, Line 232) to Figure 2, where the variable $r_{a0}$ appears in the caption, while it is later introduced on Page 9, Line 246.

**Specific comments**

Main document

Page 3, Line 79: Add comma in "... for a limited forcing, given that the …"

Page 4, Line 112: Please introduce the not SI-unit Sverdrup, for example, by a footnote or additional text.

Page 2, Line 36 – 37; Page 4, Line 107 – 198; Page 4, Line 109 – 111, Page 4, Line 118, Page 5, Line 1124 – 125: Missing citation (Madsen et al. 2022).

Page 5, Line 132: Do you mean the citation "(compare e.g. Lohmann and Ditlevsen, 2021)?

Page 5, Line 151: You may consider adding additional information: "(… continent by the ocean without floating ice shelves in Oerlemans (1981))."

Page 6, Line 156: "Thereby, the surface mass balance $a_s$ that is the difference between mass gain and mass loss is reduced and …"

Page 6, Equation 3: Is the variable $\widetilde{a}_O$ a spatial dependent variable? If so, please indicate or mention it. In addition, does this linear equation consider an increased vulnerability by a more than linear increase of the ablation zone for lowering height?

Page 6, Line 165: You may add at the end: "Furthermore, the surface mass balance equals in our setup the total mass balance."

Page 7, Equation 11: I can not find the definition of $A_i$ in the text. Please add it, even if this information is available in the supplement's table. Furthermore, does a hosing of $H = 0.2\ Sv$, as shown in Figure 2b, correspond to an additional freshwater flux of $A_i \cdot H$ ? If so, wouldn't it correspond to a hosing freshwater flux of 0.014 Sv and 0.1504 Sv into the North Atlantic and Tropical Atlantic box, respectively? Since the manuscript addresses the impact of freshwater on the AMOC stability and the conditions at and around Greenland, it is surprising that the freshwater flux may be ten times larger in the tropics than in the North Atlantic. Anyhow, please clarify this point.

Page 8, Line 217: Does the surface mass balance on the ground correspond to the surface mass balance at the sea-level?

Page 8, Line 218 – 219: You may rephrase "For a declining overturning strength $q$ of the AMOC with $H > H_{\text{ref}}$, the temperature $T_N$ in the North Atlantic box declines as well according to Eq. (4)" if applicable, e.g., "For active hosing ($H > H_{\text{ref}}$), the AMOC overturning strength $q$ declines as well as the temperature $T_N$ in the North Atlantic box is driven by Eq. (4)."

Page 8, Line 219: You may rephrase "For $d_{\text{oa}} = 0$, we recover a …" with "For $d_{\text{oa}} = 0$, we obtain a …"

Page 8, Line 220: I find the wording "independent ice sheet" confusing and misleading. Please change it.

Page 9, Line 219 – 223: You may simplify these sentences: "A unidirectional coupling is obtained by $d_{\text{oa}} = 0$, where Greenland is not exposed to any changes in the North Atlantic (Eq. 12). In addition, the freshwater flux by the Greenland Ice Sheet resulting from its mass loss (Bamber et al., 2012, 2018; Trusel et al., 2018) is added as $F_{\text{GIS}}$ to the combined freshwater into the surface North Atlantic box as:"

Page 9, Line 229: Wouldn't it be correct to state "($F_{\text{GIS}} > 0 \, \text{Sv}$)"?

Page 9, Line 231: You may write "… freshwater flux into the Atlantic Ocean, which increases the …" or?

Page, Line 245 – 247: The sentence "More specifically, the surface …. is reached (Fig. 1(b))" is not clear. Please rephrase.

Page 9, Line 245 – 247: You may change the order in this sentence: "More specifically, with a ramping $r_{\text{ao}}$, the ground surface mass balance $a_0$ decreases linearly and, once crossed the deglaction threshold $a_{0dgc}$, the ice sheet stability is not sustainable." or "More specifically, with a ramping $r_{\text{ao}}$, the ground surface mass balance $a_0$ decreases linearly, and the ice sheet becomes unstable, once crossed the deglaction threshold $a_{0dgc}$ is crossed.

Page 10, Line 256: You may rephrase "By decreasing the surface mass balance at the ground level associated … ."

Page 10, Line 256: You may rephrase to "… threshold and eventually disintegrates completely … ."

Page 10, Line 257: You may replace the beginning of the sentence: "In the following, AMOC hosing $H$ is kept constant."

Page 10, Line 260 – 265: You may rewrite it: "The freshwater volume loss resulting from the forced deglaciation of Greenland corresponds to a time-varying GIS freshwater flux $F_{\text{GIS}}$ into the North Atlantic. This time-dependent GIS freshwater flux first increases as the GIS disintegrates. Consequently, the AMOC grows, potentially overshooting its threshold (Ritchie et al., 2021), but eventually returns to $F_{\text{GIS}} = 0 \, \text{Sv}$ under otherwise constant hosing (Fig. 2(a), the AMOC trajectory approximately follows the black lines)".

Page 10, Line 267: Please consider replacing "observed" with "detected" when describing the results

of simulations since model results are not measured and turned into observed properties. Therefore, please rephrase "… to a freshwater flux as detected in previous hosing experiments … ."

Page 10, Line 269 – 272: You may rephrase " In particular, the AMOC may transition to its 'off'–state in response to the Greenland Ice Sheet disintegration, which is accompanied by a temporary over-shoot of the GIS freshwater flux threshold, resulting in an overshoot cascade (Fig. 2(c)). The increasing GIS freshwater flux puts the AMOC from the 'on' to the 'off'-state, while the AMOC does not recover after the decline of the GIS freshwater flux."

Page 10, Line 272 – 273: The following might be more appropriate: "The surface mass balance decreases substantially… , which results in a complete deglaciation of Greenland."

Page 10, Line 276: I'm unsure, but shouldn't it be: "AMOC weakening without tipping, as commonly detected in hosing … "?

Page 10, Line 278 – 279: Here is an example of avoiding unnecessary brackets: "… within 1000 years driven by a faster and stronger … ."

Page 10, Line 281: What do you think about rephrasing: "...deglaciation of Greenland, the AMOC leaves the stable 'on'-state. Rate-induced transition … ."?

Page 10, Line 284: I suggest replacing the text with "the AMOC to the changing freshwater flux by … ."

Page 10, Line 284: Please add a comma after the preposition: "Here, it is assumed … ."

Page 10, Line 285: Enclose the example by commas: " disturbances, e.g. in initial box salinities, are always .. ."

Page 10, Line 287: Add a comma for the subordinate clause at the end: "decline as studied, e.g. as scenario-dependent … ."

Page 12, Line 291 – 293: The end of the sentence is unclear; please improve the text.

Page 12, Line 294 – 295: Please extend the text to read: "of the tipping element drivers in our model."

Page 12, Line 309: add missing comma around "thus": " lower hosing values and, thus, for the AMOC … ."

Page 12, Line 312: I'm unsure, but I guess a comma is missing: "with a slow ice sheet decline, a high hosing determining the fixed … ."

Page 12, Line 314: The sentence needs to be clarified, or?

Page 12, Line 315 – 316: I would like to suggest: "an overshoot cascade changes by variations of Greenland Ice Sheet's melting patterns. More … "

Page 13, Line 340 – 342: I suggest: Here, …. of an AMOC weakening, and it may be …. surface mass balance, for a warming … ."

Page 13, Line 344: Do you mean "distinct tipping thresholds" or "different tipping thresholds"?

Page 13, Line 349: Please replace "observed" with "detected."

Page 15, Line 356 – 358: The sentence "With the AMOC tipping …. from the AMOC overturning strength" is unclear to me.

Page 15, Line 378: You may replace "ice sheet melting time" with "ice sheet disintegration time"?

Page 17, Line 389: I guess a comma is missing: "is accelerating (Shepherd et al., 2020), and its … ."

Page 17, Line 398: Since you are apparently using the British syntax predominately, replace "e.g.," with "e.g.".

Page 17, Line 404 – 406: Unclear sentence. Please rephrase.

Figure 1: Please increase the size of the hardly seen green points, which is stated in the text (Page 9, Line 235): "subcritical Hopf bifurcation at $F_{GIS}$ Hopf (indicated by green points in Fig. 2(a))."

Figure 1, caption: The introduced variable $r_{a0}$ has to be defined. Please find a way to introduce it and/or refer to the text.

Figure 3, caption: I would like to suggest the following modification to the figure caption:" Shown is the AMOC overturning strength, also taking … " (drop "now"); "… declining from pink (100 %) to grey (0 %) as indicated by the right colorbar."; "indicate the AMOC in its 'on'-state, see bottom colorbar)."

Figure 3: Please define the green arrows in the caption and drop them.

Supplement Material

Table S1: Please add missing units, e.g., "psu" for $S_0$ .

Table S1, caption: You may also define the unit Sverdrup in the caption.

Table S2: What are the missing units of the listed salinity contents? Please add.

Table S3: Since the hosing flux $H$ has the unit "Sv" in your figures (e.g., 2b), and the combined fresh-water flux according to equation (11) shall result in "Sv" as well, the unit of the parameters $A_i$ are dubious. Please check.

**Bibliography**

Madsen, M. S., S. Yang, G. Aðalgeirsdóttir, S. H. Svendsen, C. B. Rodehacke, and I M Ringgaard. 2022. "The Role of an Interactive Greenland Ice Sheet in the Coupled Climate-Ice Sheet Model EC-Earth-PISM." *Climate Dynamics*, no. 0123456789 (February). https://doi.org/10.1007/s00382-022-06184-6.

---

## Author Comment (AC1)

**Response to the comments of the reviewers for the manuscript**
**'*Rate-induced tipping cascades arising from interactions between the Greenland Ice Sheet and the Atlantic Meridional Overturning Circulation*'**

**by A.K. Klose, J.F. Donges, U. Feudel, and R. Winkelmann**

We would like to thank the reviewers for carefully reading our manuscript and for their efforts in creating their review comments. We considered their suggestions thoroughly for an adaptation of the manuscript.

In the revised version of the manuscript, we have addressed the issue of the presentation and framing of our methods and results as raised by the reviewers:

- We have reorganized Section 3, that introduces the conceptual model to capture the interaction between the Greenland Ice Sheet and the AMOC, and parts of Section 4, presenting the results.

- We have changed the naming of the 'overshoot cascade' to 'overshoot / bifurcation cascade'.

- Previously missing explanations of some parameters were added. For example, we clarified the role of the hosing as additional freshwater input to the North Atlantic through additional precipitation over the ocean and increased river runoff with warming.

We provide a point-by-point response to all comments below. The reviewers' comments are given in bold font, the authors' reply in normal font and changes to the text in italic font. To show how the proposed changes could be implemented, we attached a manuscript that highlights the suggested changes compared to the original manuscript. Deleted parts of the manuscript are crossed out and marked in red, while added parts are given in blue. Line numbers in our responses refer to this manuscript, if not stated otherwise.

We are grateful for the opportunity to further improve our manuscript and are looking forward to your feedback.

Sincerely yours,

Ann Kristin Klose, Jonathan F. Donges, Ulrike Feudel, and Ricarda Winkelmann

*Reviewer Comment 1*

**This paper by Klose et al examines how tipping cascades can emerge through interactions between the Greenland Ice Sheet and the Atlantic Meridional Overturning Circulation. They use a simple, but physically motivated, model of these systems to show that tipping cascades can emerge due to bifurcations and also due to rate induced tipping. I thought the analysis of the results was good, but the paper could be improved with regard to its presentation and framing.**

**I organise the review as follows: I give some broad comments about the paper, followed by some specific points.**

We are grateful for the overall positive evaluation of our analysis on the dynamics of the interacting Greenland Ice Sheet and AMOC. We thank the reviewer for carefully reading our manuscript and providing helpful comments to improve our manuscript, in particular, with regard to the presentation of our results.

**Broad Comments:**

**What is the relation of this work to that of Sinet 2023, which also examines a simple model of the interaction of the AMOC with ice sheets?**

Sinet et al. (2023) studies the dynamics that may arise in a chain of tipping elements consisting of the AMOC and the ice sheets in Greenland and Antarctica, by means of coupled physically-motivated conceptual models (of similar complexity as the model presented in our manuscript). Their approach captures the freshwater fluxes into the ocean (thereby affecting the AMOC) that arise from a disintegration of the Greenland and West Antarctic Ice Sheet. In addition, temperature changes in the southern Atlantic Ocean are related to the ice dynamics in Antarctica by the depth integrated ice viscosity parameter.

The potentially stabilizing effect of an AMOC decline on the Greenland Ice Sheet given a cooling in the Northern Hemisphere is not included in Sinet et al. (2023), but discussed as a relevant next step in future research. While not taking into account the West Antarctic Ice Sheet in our study of interacting tipping dynamics, we here additionally include this potentially stabilizing effect via relative temperature changes and thus assess the positive-negative feedback loop between the Greenland Ice Sheet and the AMOC.

Eventually, both approaches may be combined towards a network of coupled tipping elements, to explore the dynamics of the interacting tipping element network described in Wunderling et al. (2021) by means of physically-motivated conceptual models to further inform fully-coupled Earth system modelling.

In the Introduction of the revised manuscript (lines 58-62), we have clarified the relation to the work by Sinet et al. (2023) as:

> *The dynamics of the AMOC and ice sheets in Greenland and West Antarctica as a chain of tipping elements was assessed by Sinet et al. (2023). Here, the AMOC may be stabilized by a disintegration of the West Antarctic Ice Sheet, thereby potentially hindering cascading tipping in the climate system. The stabilizing effect of a net-cooling around Greenland with an AMOC weakening is not included in the modelling approach of Sinet et al. (2023).*

**The paper distinguishes between overshoot and rate induced cascades. I think it would be better to call overshoot cascades "bifurcation cascades" instead, as this reflects their tipping mechanism. The term overshoot is connected to notions of reducing the forcing back to a safe level.**

We thank the reviewer for this remark and suggestion. Our notion of 'overshoot cascade' is motivated by the detected dynamics of the Greenland Ice Sheet and the AMOC in our model: The AMOC collapses with the crossing of a tipping threshold due to increased freshwater flux from the Greenland Ice Sheet. This tipping threshold may, indeed, be associated with a bifurcation (corresponding to bifurcation-induced tipping). With the complete loss of the Greenland Ice Sheet, the freshwater flux declines again below this tipping threshold. As such, the tipping threshold is only crossed for some time ('overshooted'). This 'overshoot' is, however, not safe, in contrast to dynamics discussed recently as 'safe overhoots' in the context of tipping elements (e.g., Ritchie et al., 2019, 2021).

To reflect both aspects, we have changed the wording to 'overshoot / bifurcation cascade' in the revised manuscript.

**How do the rates and magnitudes of the changes in the control parameter relate to the rates and magnitudes of changes in observations and future projections under different climate scenarios?**

This is an interesting question, and we thank the reviewer for motivating the following discussion:

Overall, the freshwater fluxes from the Greenland Ice Sheet resulting from the different forcing scenarios in our flowline model are within the range of future projections with fully-dynamic ice-sheet models. We here focus on comparing our results to long-term projections of Greenland mass changes that show the loss of the entire ice sheet.

For example, Van Breedam et al. (2020) assess the future evolution of the ice sheet on Greenland over the next 10 000 years for a range of scenarios differing in the atmospheric carbon dioxide forcing. These scenarios extend ECP pathways and result in the loss of the entire Greenland Ice Sheet over varying timescales ranging from about 10 000 years (MMCP2.6) to about 2500 years (MMCP8.5). This may give rise to a freshwater flux on the order of 0.009 Sv to 0.03 Sv. Note that we here (and in the following approximations) assume that the total ice volume on Greenland (of about $2.99*10^6$ km³, based on Morlighem et al., 2017) is lost with a constant rate over time. Given the non-linear behavior of the Greenland Ice Sheet, freshwater fluxes could be higher over certain time periods during the deglaciation, e.g. between 0.03 – 0.05 Sv for MMCP8.5 as reported in Van Breedam et al. (2020).

Robinson et al. (2012) find an even faster deglaciation of Greenland within 2000 years in their experiments for a warming of 8°C, equivalent to a (constant) freshwater flux of 0.04 Sv. This is comparable to the maximum freshwater flux from ice loss in Greenland during its deglaciation within 3000 years in the overshoot / bifurcation cascade, presented in Figure 2(c) of the original manuscript.

Finally, in Aschwanden et al. (2019), Greenland likely becomes ice-free by the end of the millennium (deglaciation within < 1000 years) under the higher-emission pathway RCP8.5, giving rise to a (constant) freshwater flux of 0.09 Sv. A relatively fast deglaciation of Greenland is necessary for a rate-induced cascade; in the example given in Figure 2(d) in the original

manuscript within 1000 years. This is consistent with Greenland becoming ice-free under RCP8.5 in Aschwanden et al. (2019).

Observed rates of ice sheet mass loss amounted to 35 Gt yr$^{-1}$ in 1992-1996 (equivalent to a freshwater flux of about 0.001 Sv) and increased to 257 Gt yr$^{-1}$ in 2017-2020 (equivalent to approximately 0.008 Sv). The observed rates of mass loss from Greenland are thus still smaller than projected changes by an order of magnitude.

We have included a short paragraph comparing the GIS deglaciation timescales within the different tipping cascades in our conceptual model with previous projections by means of fully-dynamic ice-sheet models (Robinson et al. 2012; Aschwanden et al., 2019) in the revised manuscript (lines 311-314 and lines 320-322).

> *The surface mass balance is decreased substantially beyond the deglaciation threshold to $a_{0max}$ =-3.0 m a$^{-1}$ within about 3000 years, which results in a complete deglaciation of Greenland in this time period. This deglaciation timescale and the resulting freshwater flux is of a comparable order of magnitude as determined for the ice-sheet collapse given a constant regional summer temperature rise of 8 °C in Greenland in a fully-dynamic ice-sheet model (Robinson et al., 2012).*

> *A faster and stronger decrease of the surface mass balance may drive a more extreme collapse of the GIS within about 1000 years, which is comparable to Greenland becoming ice-free until the end of the millennium under the higher-emission pathway RCP8.5 in Aschwanden et al. (2019).*

**I often think of hosing as being related to the melting of the Greenland ice sheet, yet here it is included as an additional process. What process is this capturing and what are plausible values of H?**

We thank the reviewer for this comment and agree that hosing is often associated with Greenland Ice Sheet melting.

In our modelling approach, the freshwater flux from the Greenland Ice Sheet is derived from the physically-motivated flowline model. We consider the hosing H as an additional surface freshwater input into the North Atlantic through increased river runoff and precipitation over the ocean with a warmer climate (e.g., Fox-Kemper et al., 2021). The hosing pattern in the global ocean box model is dictated by the multiplicative factors $A_i$ (compare our response to related comments). These are combined with 'baseline' surface freshwater fluxes directly obtained from simulations under pre-industrial (1x$CO_2$) conditions with the AOGCM FAMOUS (Wood et al., 2019). Note that no Greenland Ice Sheet and hence no meltwater fluxes are included in FAMOUS experiments.

As noted by Reviewer 2, a change in the hosing H in our experiments brings the AMOC closer to its critical (hosing) threshold (compare Figure 2(a) in the original manuscript), making the AMOC more susceptible to additional freshwater from Greenland. Eventually, the interplay of all of these freshwater fluxes determines a propagation of tipping from the ice sheet on Greenland to the AMOC in our model.

In the revised manuscript, we have added a definition of the hosing H and its role in our experiments, also taking into account a related comment from Reviewer 2. In lines 223-225 of the revised manuscript, we have added the following explanation:

*These additional surface freshwater fluxes based on the hosing H are here considered as increased river runoff and precipitation over the ocean into the North Atlantic with a warmer climate.*

**For the first part of the paper the parameter d_{oa}, which controls the thermal coupling of the AMOC and GIS is set to zero. Would it not be clearer to maintain it at the non-zero level throughout the paper, particularly as the interaction between the tipping elements is a focus of the paper.**

We here aim at exploring the dynamics and, in particular, the risk of cascading tipping dynamics emerging from the interactions of GIS and AMOC in a positive-negative feedback loop. To distinguish the effects of freshwater fluxes into the North Atlantic on the one hand and a relative cooling around Greenland on the other hand on the overall dynamics of these tipping elements, we have decided to – step-by-step – add the different interaction mechanisms forming this positive-negative feedback loop in our analysis. This also allows a qualitative comparison of the AMOC tipping behaviour in response to a deglaciation of Greenland detected in our conceptual, physically-motivated approach to previous hosing experiments. These hosing experiments neglect the effect from a declining AMOC on the Greenland Ice Sheet and imposed freshwater fluxes.

Following the suggestion of Reviewer 2, we have modified the title of the section that neglects the thermal coupling of the AMOC and GIS in the revised manuscript, to clearly indicate the considered interactions in this section.

**The model is described as process based, but I don't think that's quite right. For example the box model of the AMOC does not represent real processes in the system. I think this modelling approach is fine, but it is probably fairer to describe the model as 'physically motivated' for example.**

We thank the reviewer for this suggestion. Following the suggestion, the model is described as 'physically-motivated' in the revised manuscript.

**Specific comments:**

**Is line 51 consistent with line 39?**

We agree with the reviewer that the statements in line 39 and line 51 of the original manuscript in their current formulation can be misleading.

In line 39 of the original manuscript we state that the effect of the interaction between the Greenland Ice Sheet and the AMOC in a positive-negative feedback loop on the overall stability is largely unknown. This does not exclude some knowledge, such as the response of the AMOC to an idealized freshwater flux determined by models of varying complexity, as outlined in Section 2 of the original manuscript.

In addition, Wunderling et al. (2021) assessed the stability of a network of tipping elements, taking into account their interaction structure, as described in line 51 in the original manuscript. Their assessment is based on combining conceptual representations of tipping elements and existing knowledge on uncertainties in the critical temperature thresholds as well as the strengths of the interaction between tipping elements in a risk analysis approach. The conceptual representation of tipping elements is motivated by e.g. results from processedbased models and paleoclimate evidence. Their approach allows to explore the qualitative role of known tipping element interactions in a systematic way.

We have clarified this inconsistency in the revised manuscript and thank the reviewer for pointing it out. More specifically, the related paragraphs in the revised manuscript (lines 40-41 and lines 52-55) now read as:

> *There is still a knowledge gap of the effect of this positive-negative feedback loop on the overall stability of the coupled system of climatic tipping elements.*

> *The interactions between these four key climate tipping elements tend to be overall destabilizing under ongoing warming as suggested by integrating expert knowledge and including uncertainties of critical temperature thresholds and interaction strengths into a risk analysis approach for these interacting tipping elements.*

**Lines 59,91 and 405 are too strong. The IPCC has low confidence in historical reconstructions of the AMOC, and the findings of Boers and Rypdal are consistent with the approach of a critical threshold but other explanations are possible.**

We agree with the reviewer that AMOC reconstructions and thus trend estimates are associated with high uncertainties. Following the comments from all reviewers, we have reformulated lines 59, 91, and 405 of the original manuscript on the evolution of the AMOC and the Greenland Ice Sheet and the possible approach of tipping points.

In the revised manuscript (lines 63-66, 98-100 and 467-471), the related paragraphs now read as:

> *Significant changes of both systems are observed at present with an acceleration of GIS mass loss (Shepherd et al., 2020; Trusel et al., 2018) as well as a weakening of the AMOC (Caesar et al., 2018), though AMOC reconstructions are associated with high uncertainties (Moffa-Sánchez et al., 2019). There is limited evidence that these changes may be related to the approach of a critical threshold with ongoing global warming (Boers and Rypdal, 2021; Boers, 2021; Van Westen et al., 2024).*

> *Based on early warning signals the proximity of a critical threshold in west Greenland (Boers and Rypdal, 2021) and a potential loss of stability of the current strong AMOC mode (Boers, 2021; Van Westen et al., 2024) have been suggested.*

> *A decline of 15 % in the strength of the overturning circulation since the mid-twentieth century is found in the observed sea-surface temperature trend (Caesar et al., 2018) and it is suggested that the current AMOC state might lose stability (Boers, 2021; Van Westen et al., 2024).*

**Line 190: What is meant by analogously here?**

We agree with the reviewer that this formulation in the revised manuscript may have been confusing. The salinity dynamics in each box is governed by salt conservation, which guides the formulation of Eq. 6-9. This is also the case for a negative overturning strength $q < 0$, and

the governing equations can be formulated in analogy to Eq. 6-9. In the revised manuscript, we have added the following sentence (lines 210-211):

> *A second set of equations for the salinity evolution in each box in the case q < 0 can also be formulated based on salt conservation.*

**Line 199: What is A_i?**

We thank the reviewer for pointing out the missing explanation of the parameter $A_i$ in the original manuscript.

The parameters $A_i$ in the global ocean box model are multiplicative factors for the hosing in the respective ocean boxes. That is, the 'uniform' hosing parameter H is scaled for each of the boxes, according to these factors. The box model parameters are calibrated based on GCM experiments (Wood et al., 2019). In this calibration, the values for the factors $A_i$ are chosen to match the total freshwater flux $A_i*H$ into each of the boxes as in the GCM when imposing hosing. Here, we have adopted the parameters obtained in a calibration with FAMOUS, following Wood et al. (2019).

The following explanation has been added in the revised manuscript (lines 218-220):

> *Here, $F_{i0}$ are considered as baseline surface fluxes of the respective ocean boxes under pre-industrial conditions, and $A_i$ are multiplicative factors distributing additional surface freshwater fluxes across the boxes based on the hosing H (Wood et al., 2019).*

**Line 200: Why should fluxes be balanced? If water was added in the North Atlantic box I wouldn't a priori expect it to be removed elsewhere.**

This is a fair point and we thank the reviewer for this remark.

Outside the modelling world, salinity changes arise by changing the volume of the ocean boxes with freshwater fluxes such as from the Greenland Ice Sheet, while the overall salt content of the ocean is constant.

In the global ocean box model, we apply salinity fluxes that change the salt content of the ocean, while keeping the volume constant, and as in previous box modelling approaches, the volume of the ocean boxes is not changed with additional freshwater input. However, this may be a next step in future research. To adopt this approach, fluxes are balanced.

**Line 232 "As exemplarily indicated" should read "As indicated"**

Done.

**Line 265 "For exemplary" should read something like "For different"**

This is reformulated in the revised manuscript.

**Lines 296 to 305 Does figure 4 refer to figure 3?**

Yes. We have corrected this in the revised manuscript.

**Line 355 "As exemplarily shown" should read "As shown"**

Done.

**Figure 4 a,b,d,e could have a zoomed in x-axis, possibly each with a different scale as most of the interesting dynamics happen early in the simulation.**

We thank the reviewer for this suggestion. We have added a zoom-in for Figure 4 in the revised manuscript.

**Line 386: "Pending" should read something like "yet"**

Done.

*Reviewer Comment 2*

**This manuscript investigates the different manners in which cascade tipping between the Greenland ice sheet and the Atlantic meridional overturning circulation can take place, depending on the rate of forcing resulting from Greenland melting and the existence of a negative feedback of the reduced AMOC on Greenland temperature. The manuscript is in general well written and the results are interesting, although limited by the simplicity of the model used. I think it deserves to be published once a few issues are solved. Most notably, the writing is a bit confusing in some points and the Results section is a bit entangled, so I would recommend restructuring it somewhat.**

We are thankful for this overall positive evaluation and are happy that the reviewer considers our results as interesting. We would like to thank the reviewer for the comments which helped to improve the manuscript and, in particular, its structure.

**Specific issues:**

**Line 12: replace "breaching" by "surpassing"**

Done.

**Line 20: references should be included to refer to the melt-elevation feedback and the positive salinity advection feedback (e.g. Robinson et al. 2012 and Rahmstorf 1996, both of which are already included in the Reference list)**

We have added these references to refer to the melt-elevation feedback and the salt-advection feedback in the revised manuscript.

**Lines 30, 38: "The Greenland Ice Sheet and Atlantic Meridional Overturning Circulation are strongly linked via freshwater fluxes into the North Atlantic originating from a melting GIS on the one hand, and via a relative cooling around Greenland with a slowdown of the AMOC on the other hand (Kriegler et al., 2009; Sinet et al., 2023)." This statement refers to studies consisting of expert elicitation and a conceptual model. I would compel the authors to try to find more physically-based support in the literature (actually these are provided in lines 36-37).**

We thank the reviewer for this important remark, and have added the following references that support these interaction mechanisms in the revised manuscript:

> *Bamber et al. (2012), Bamber et al. (2018), Vellinga et al. (2002), Vellinga et al. (2008), Jackson et al. (2015)*

**Line 35: the question as to whether the AMOC is already decreasing is controversial, I would recommend having a more balanced discussion. This does not preclude that projections all indicate an AMOC reduction in the future, so the question is relevant.**

We agree with the reviewer that AMOC reconstructions and thus trend estimates are associated with high uncertainties.

In lines 32-37 of the original manuscript, we describe the proposed positive-negative feedback loop and related physical mechanisms in the suggested interaction of the Greenland Ice Sheet and the AMOC. The potential weakening effect of freshwater fluxes from the (future) decline of the Greenland Ice Sheet is one of these mechanisms. In the revised manuscript (lines 34-37), we reformulated this paragraph, suggesting that the AMOC may weaken by increasing mass loss from Greenland and added an additional reference referring to a modelled future AMOC slowdown driven by meltwater from the Greenland Ice Sheet (Golledge et al. 2019):

> *More specifically, the increasing mass loss of the GIS (Shepherd et al., 2020; Mouginot et al., 2019; Van den Broeke et al., 2016) results in a freshwater input to the North Atlantic (Bamber et al., 2012, 2018; Trusel et al., 2018), which may weaken the AMOC by decreasing sea water density and thereby weakening deep water formation (Caesar et al., 2018; Rahmstorf et al., 2015; Golledge et al., 2019).*

Related to this reviewer comment and following the comments from all reviewers, we have also reformulated lines 59, 91, and 405 of the original manuscript on the evolution of the AMOC and the Greenland Ice Sheet and the possible approach of tipping points.

In the revised manuscript (lines 63-66, 98-100 and 467-471), the related paragraphs now read as:

> *Significant changes of both systems are observed at present with an acceleration of GIS mass loss (Shepherd et al., 2020; Trusel et al., 2018) as well as a weakening of the AMOC (Caesar et al., 2018), though AMOC reconstructions are associated with high uncertainties (Moffa-Sánchez et al., 2019). There is limited evidence that these changes may be related to the approach of a critical threshold with ongoing global warming (Boers and Rypdal, 2021; Boers, 2021; Van Westen et al., 2024).*

> *Based on early warning signals the proximity of a critical threshold in west Greenland (Boers and Rypdal, 2021) and a potential loss of stability of the current strong AMOC mode (Boers, 2021; Van Westen et al., 2024) have been suggested.*

> *A decline of 15 % in the strength of the overturning circulation since the mid-twentieth century is found in the observed sea-surface temperature trend (Caesar et al., 2018) and it is suggested that the current AMOC state might lose stability (Boers, 2021; Van Westen et al., 2024).*

**Line 91: same as above, the question of whether the AMOC is approaching a tipping point is controversial.**

Please compare our response to the previous comment.

**Lines 101, 132: suppress "compare"**

Done.

**Lines 133: Besides Robinson et al. you could refer also to the recent study by Bochown et al:**

**Bochow, N., Poltronieri, A., Robinson, A. l. Overshooting the critical threshold for the Greenland ice sheet. Nature 622, 528–536 (2023).** https://doi.org/10.1038/s41586-023-06503-9

We have included Bochow et al. (2023) as an additional reference in the revised manuscript.

**Line 143: insert commas before and after "based on the shallow-ice approximation (Hutter 1983)"**

Done.

**Lines 148-149: consider rewriting this as "the ice flux F and the mass balance at the surface as (first and second term on the right hand side of Eq (1), respectively".**

Done.

**Line 150: rewrite as "from x = -L to x = L" or "between x = - L and x = L"**

Done.

**Line 152: insert commas before and after "Eq(1)-(2)"**

Done.

**Lines 167, 170: I would write these inequalities in the inverse sense (e.g. a0 > a0gc > 0 for the first one)**

Done.

**Line 173: replace "to include" by "including"**

Done.

**Equation 11: explain Fi0 and Ai**

We thank the reviewer for pointing out the missing explanation of the parameters $A_i$ and $F_{i0}$ in the original manuscript.

The parameters $A_i$ in the global ocean box model are multiplicative factors for the hosing in the respective ocean boxes. That is, the 'uniform' hosing parameter H is scaled for each of the boxes, according to these factors. The box model parameters are calibrated based on GCM experiments (Wood et al., 2019). In this calibration, the values for the factors $A_i$ are chosen to match the total freshwater flux $A_i*H$ into each of the boxes as in the GCM when imposing hosing. Here, we have adopted the parameters obtained in a calibration with FAMOUS, following Wood et al. (2019).

$F_{i0}$ are the 'baseline' surface fluxes of the ocean boxes under pre-industrial conditions without additional hosing, and are likewise obtained from the GCM experiments.

The following explanation is added in the revised manuscript (lines 218-220):

*Here, $F_{i0}$ are considered as baseline surface fluxes of the respective ocean boxes pre-industrial conditions, and $A_i$ are multiplicative factors distributing additional surface freshwater fluxes across the boxes based on the hosing H (Wood et al., 2019).*

**Line 203: introducing freshwater as salinity fluxes was a practice done in old OGCMs using the rigid lid approximation.**

Yes, this is correct. We have added this information in the revised manuscript (lines 225-228) as follows:

*Note that freshwater fluxes are introduced as virtual salinity fluxes based on a reference salinity as in previous ocean box models, e.g., (Rahmstorf 1996; Lucarini and Stone, 2005) and likewise in some GCMs, e.g., (Swingedouw et al., 2013; Yin et al., 2010, Rahmstorf 1996), that often apply a rigid lid approximation.*

**Line 212: suppress new paragraph here**

Done.

**Lines 209-232: This is not really part of the Results section, but should be part of section 3.**

We agree with the reviewer that lines 209-232 in the original manuscript may rather be considered as part of Section 3, describing the modelling approach. Following the suggestions of all reviewers, we have reorganized Section 3 and 4 in the revised manuscript:

In particular, the first part of Section 4.1 in the original manuscript (lines 209-232) on modelling the interaction between the Greenland Ice Sheet and the AMOC is shifted to Section 3 as additional Section 3.3 in the revised manuscript.

Lines 232-241 in the original manuscript describing the bifurcation structure of the AMOC with varying freshwater flux remains in Section 4 as Section 4.1, and is titled 'AMOC bifurcation structure for varying freshwater fluxes' in the revised manuscript.

The final part of Section 4.1 in the original manuscript (lines 242-248) describing the transient experiments for exploring the tipping dynamics of the interaction Greenland Ice Sheet and AMOC remains in Section 4 to introduce Section 4.2 on 'Tipping cascades between GIS and AMOC without negative feedback'. This is in line with other comments by Reviewer 2 and 3. In particular, we hope that, by introducing these sections with a description of the applied scenarios and the definition of the related parameters $r_{a0}$ and $a_{0max}$, our experiments become clearer.

**Lines 232-241: Here there is a discussion with references to Figure 2a which corresponds to the case without the negative feedback; therefore I think it should appear later on, within 4.2. Also, how do you know the bifurcation is a Hopf bifurcation? If this is based on the analysis by Alkhayoun et al. (2019) it should be made more clear, otherwise the reader thinks they have missed something.**

Following the suggestions of all reviewers, we have reorganized Section 3 and 4 in the revised manuscript. Please compare our response to the previous comment for a more detailed description of the related adjustments.

We agree that a full description of the (uncoupled) global ocean box model bifurcation structure, as presented in Alkhayuon et al. (2019), is missing in the original manuscript. In the revised manuscript, we have therefore introduced the following short paragraph in the beginning of Section 4.1 in the revised manuscript (lines 256-259):

> *Depending on the hosing H, a strong 'on' and a weak 'off' AMOC configuration may coexist as stable states in this global ocean box model (Fig.2(a), indicated in blue). The AMOC 'on'-state loses stability via a subcritical Hopf bifurcation upon crossing the hosing threshold $H_{Hopf}$, as shown by Alkhayuon et al. (2019). It eventually disappears when it meets the separating saddle (Fig.2(a), indicated as dashed blue) in a fold.*

**Equation 13: I understand that the role of H here is simply to bring the system closer to its critical threshold in order to study the sensitivity of the response to the starting point, right? If so I would try to say it more clearly.**

We thank the reviewer for asking for a clarification on the role of the hosing H.

As correctly stated by the reviewer, a change in the hosing H in our experiments brings the AMOC closer to its critical (hosing) threshold (compare Figure 2(a) in the original manuscript), making the AMOC more susceptible to additional freshwater from Greenland. To clarify the role of the hosing H, we have added the following explanation in the revised manuscript (lines 289-290) when describing our experiments:

> *In particular, by increasing the hosing H the AMOC is brought closer to its critical (hosing) threshold, changing its susceptibility to an additional freshwater flux from Greenland.*

We consider the hosing H as surface freshwater input into the North Atlantic through increased river runoff and precipitation over the ocean with a warmer climate (e.g., Fox-Kemper et al., 2021). These are combined with the freshwater flux from the Greenland Ice Sheet, derived from the physically-motivated flowline model in our modelling approach.

In line with a related comment by Reviewer 1, we have also added a definition of the hosing H in the revised manuscript. In lines 223-225 of the revised manuscript, we have added the following explanation:

> *These additional surface freshwater fluxes based on the hosing H are here considered as increased river runoff and precipitation over the ocean into the North Atlantic with a warmer climate.*

**Line 226: I would suppress "It eventually acts as a virtual salinity flux, while assuming a constant ocean volume (compare Section 3.2)", no need to state again.**

We agree with the reviewer that the use of virtual salinity fluxes in the ocean box model was already introduced in the model description (Section 3.2). To clarify that this also applies to

the freshwater flux from Greenland, we would like to keep this sentence in the revised manuscript.

**Lines 240-241: I would rewrite "it is eventually attributed to a time–dependent decline of the GIS and in fact turns into a state variable in transient experiments" as "it is actually a state variable in transient experiments that represents the freshwater forcing into the North Atlantic due to the time–dependent decline of the GIS"**

Done.

**Lines 242-248: I would move this paragraph to the beginning of the next section, because it contains important information concerning the experimental setup followed.**

Thanks for this suggestion. We have moved this paragraph to introduce the Section 4.2 on 'Tipping cascades between GIS and AMOC without negative feedback' of the revised manuscript. Please compare our response to the previous comment for a more detailed description of adjustments related to this section in the revised manuscript.

**Line 249: I think you need to include "without the negative feedback" in the title**

We thank the reviewer for this suggestion. We have added 'without the negative feedback' in the Section title in the revised manuscript.

**Line 251: "complementing previous freshwater experiments" - which??**

We have added the following references on previous freshwater hosing experiments in the revised manuscript:

> *Hu et al. 2009; Jungclaus et al 2006; Stouffer et al. 2006; Swingedouw et al. 2013;*
> *Swingedouw et al. 2015; Rahmstorf et al. 1995*

**Line 258: I would suppress " The AMOC hosing H is kept constant", there is no need to say this again.**

Done.

**Line 264: FGIS = 0 Sv is reached one the GIS has completely melted, right? If so I would say so.**

This is correct. We have added this information in the revised manuscript.

**Line 265: "For exemplary melting patterns…": this wording is a bit confusing. When one reads "melting patterns" it reminds of spatial patterns which have nothing to do with this study. "Exemplary" also sounds strange. Why not just say that there are two possible modes of cascade tipping and give one example for each?**

We agree with the reviewer that the use of 'patterns' suggests that the freshwater fluxes from the Greenland Ice Sheet differ in their spatial dimension, which we cannot capture with our

flowline modelling approach. As the relevant characteristic of the decline of the Greenland Ice Sheet in our model is its timescale, we have changed the phrase 'melting pattern' to 'disintegration time' in the revised manuscript. This is also motivated by a suggestion of Reviewer 3.

In the revised manuscript (lines 303-304), it now reads as:

*Depending on the disintegration time of the GIS and positions of the AMOC relative to its hosing threshold we can identify different types of cascading tipping of the GIS and the AMOC.*

**Line 272: I think this part would be better understood with a bit more information on the parameters H, rao and a0max, which currently are only shown in the caption of Figure 2. The same goes for the discussion in lines 278-285.**

We agree that the definition of the parameters $H$, $r_{a0}$ and $a_{0max}$ could be improved. The evolution of the surface mass balance on the ground $a_0$, determined by the parameters $r_{a0}$ and $a_{0max}$, is described in lines 245-248 in the original manuscript. In the revised manuscript, we have adjusted the introduction of these parameters. In particular, this paragraph is moved to introduce the Section 4.2 on 'Tipping cascades between GIS and AMOC without negative feedback'.

**Lines 298, 304: I think this should be Figure 3 rather than Figure 4.**

We thank the reviewer for pointing this out. This is corrected in the revised manuscript.

**Lines 304-305: I would move this sentence to the end of the discussion rather than anticipating the existence of rate-induced AMOC collapse already.**

This sentence is shifted to the end of this section in the revised manuscript (lines 376-377).

**Line 316: same issue with "melting patterns" as above.**

Please compare our response to the reviewer's comment above.

**Line 332-334: I would delete "The negative feedback…"; this is clear at this point.**

Done.

**Lines 342-346: where is this to be inferred? Also, would it be possible to give a specific example, with values that can be read from figure 4?**

Theoretical work on the cascading dynamics of interacting tipping elements, e.g. linearly coupled as driving (or 'master') and responding system (e.g., Klose et al. 2020), has shown that the critical transition of the following system may occur at lower or higher values of the control parameter (e.g. temperature) compared to the intrinsic tipping point (that is, the tipping point without any coupling) due to the coupling. In other words, tipping occurs when crossing effective tipping points of the responding system. In the case of such a linear coupling, these

effective tipping points depend on the direction of coupling and the state of the leading tipping element.

This theoretical basis can be linked to the Greenland Ice Sheet and the AMOC, when only the coupling via relative temperature changes (negative feedback) would be considered (unidirectional coupling). Here, the AMOC could be considered as a driving system, while the ice sheet on Greenland would correspond to the responding system.

These effective deglaciation thresholds of Greenland when considering the negative feedback via relative temperature changes on the ice sheet are displayed by the (vertical) solid black lines in Figure 4(c) and (f). They depend on the state of the AMOC in terms of the temperature of the 'on'- and 'off'-state. When comparing to the dashed grey line in this figure, indicating the deglaciation threshold when neglecting this feedback, the effect of the coupling becomes visible. The existence of these two different deglaciation thresholds is the motivation for studying different scenarios for the surface mass balance evolution and their effect of the coupled GIS-AMOC dynamics.

In the revised manuscript, we have reformulated the related paragraph, adding a more detailed explanation of effective tipping points for interacting tipping elements (lines 389-403) and references to Figure 4 in the description of these deglaciation thresholds:

> *Considering this negative feedback, the intrinsic tipping point of the Greenland Ice Sheet (that is, the critical threshold of the Greenland Ice Sheet without any coupling, compare Fig. 4(c) and (f), dashed grey; Klose et al. 2020) is replaced by two separate effective deglaciation thresholds $a_{0dgc}^{(1)}$ and $a_{0dgc}^{(2)}$ of the GIS (Fig. 4(c) and (f), solid black), depending on the state of the AMOC. This is based on the theoretical foundations of cascading dynamics for linearly coupled driving (or 'master') and following tipping elements, formulated in Klose et al. (2020): Interactions shift the critical threshold of a responding system beyond which tipping is expected to lower or higher values compared to the intrinsic tipping point depending on the direction of coupling and the state of the driving tipping element, giving rise to effective tipping point(s) of the responding system. Here, when considering the stabilizing effect of an AMOC weakening on the ice sheet (Eq. (12)), the AMOC could be considered as the driving system, while the ice sheet on Greenland would represent the responding system. Based on Eq. (12), that linearly relates the AMOC state in terms of the North Atlantic box temperature and the GIS surface mass balance, two deglaciation thresholds $a_{0dgc}^{(1)}$ and $a_{0dgc}^{(2)}$ may then be crossed with a decreasing surface mass balance in a warming climate (Fettweis et al. 2013): For $a_0 < a_{0dgc}^{(1)}$ a complete melting of the GIS is obtained given that the AMOC resides and remains in its 'on'--state. Given that the AMOC resides in its 'off'--state, the GIS melts down completely for $a_0 < a_{0dgc}^{(2)}$.*

**Lines 352-354: " Hence, for a strong surface mass balance decrease the tipping outcomes in terms of the final GIS and AMOC states when neglecting the negative feedback via the temperature are qualitatively resembled (Fig. 4(c))" : The meaning of this sentence is unclear to me.**

We here compare the possible tipping outcomes of the interacting GIS and AMOC including the negative feedback (A) to the dynamic regimes detected when neglecting this negative

feedback (B) in Section 4.2.2 of the original manuscript. For a strong surface mass balance decrease, there is no qualitative difference between both cases (A) and (B). We have reformulated this sentence in the revised manuscript, and hope it got clearer. In the revised manuscript (lines 411-414), this sentence now reads as:

> *Hence, for a strong surface mass balance decrease, the potential dynamic regimes with Greenland becoming ice-free as well as a strong or collapsed AMOC depending on the hosing (Fig. 4(c)) is comparable to the dynamics detected when neglecting the negative feedback (Fig. 3).*

**General:**

**Acronyms for the Greenland ice sheet (GIS) and the Atlantic meridional overturning circulation (AMOC) are introduced but not systematically used subsequently, this should be corrected (e.g. lines 30, 41, 48, 86, 94, 97, etc)**

For better readability of the manuscript, we decided to not use the acronyms across the entire manuscript but to have a balanced use of acronyms and full text.

**This might be a matter of taste, but I feel the text abuses a bit of using parentheses to provide explanations; I would recommend replacing those by inserting the sentence between commas. e.g. in line 5 I would replace  "(with a destabilizing effect on the AMOC due to ice loss and subsequent freshwater flux into the North Atlantic as well as a 5 stabilizing effect of a net–cooling around Greenland with an AMOC weakening) by  ", with a destabilizing effect on the AMOC due to ice loss and subsequent freshwater flux into the North Atlantic as well as a 5 stabilizing effect of a net–cooling around Greenland with an AMOC weakening, "**

**The same applies to lines 8, 19, 152, etc.**

Following the comments from all reviewers, we have included explanations in brackets either in the main sentence (separated by commas) or integrated them as additional sentences, where applicable.

**Figures**

**Figure 1b: please make this panel make larger**

Done.

**Figure 2:**

1. **Green lines indicating the thresholds are barely seeable**
2. **and d) should have the same scale for the x axis**

We thank the reviewer for the suggestions on improving Figure 2. We have increased the width of the green lines indicating the thresholds and adjusted the scales of the x-axis in the revised manuscript.

**Review of Klose, Donges, Feudel, and Winkelmann: "Rate–induced tipping cascades arising from interactions between the Greenland Ice Sheet and the Atlantic Meridional Overturning Circulation." (Earth System Dynamics, Paper: esd-2023-20)**

**The manuscript of Klose and others presents the results of a conceptional model system where a global ocean interacts with the Greenland Ice Sheet (GrIS). They allow tipping cascades of the Atlantic Meridional Overturning Circulation (AMOC) stability by freshwater released into the North Atlantic. Besides a hosing flux, the meltwater contribution of disintegrating GrIS disturbs the stability of the AMOC and, via a potential feedback loop, controls Greenland's meltwater contribution. The authors access overshot and rate-induced tipping cascades in this highly idealized system.**

**The first conceptional model renders the global ocean by hydrographic properties in five boxes: Two dedicated Atlantic surface ocean regions (North Atlantic: N, Tropical Atlantic: T) beside surface boxes for the Southern Ocean (S) and the Indo-Pacific Ocean (IP). Those surface boxes communicate with a global bottom ocean box (B). In this model, the density difference between the Northern Atlantic and Southern Ocean drives the AMOC, which controls the North Atlantic temperature and determines the temporal salt flux between different boxes (Eq. 6 – 10).**

**The other conceptional model represents the Greenland Ice Sheet by a flowline model solving the shallow ice approximation. It has a (half) width of 1000 km, which equals approximately its actual latitudinal extent. The ice loss and gain are exclusively described by the surface mass balance (SMB), where the Lapse rate effect constitutes the melt elevation feedback.**

**The coupling between these two models is unidirectional or bidirectional. The unidirectional coupling considers only the meltwater flux of a shrinking GrIS into the Atlantic Ocean. This meltwater flux plus an additional freshwater hosing flux decreases the salinity in the Atlantic. In the bidirectional setup, the North Atlantic temperature feeds back on Greenland's surface mass balance.**

**The authors detect overshoot and rate-induced cascading tipping in their model system with a focus on the GrIS and the AMOC. These cascades are analyzed in the uni- and bidirectional setups where the bidirectional coupling considers the feedback loop between the North Atlantic temperature and the ice loss. This thermal coupling stabilizes the AMOC and the ice sheet because an enhanced meltwater flux reduces, via the density difference, the AMOC strength, lowering the North Atlantic temperatures, which, ultimately, damps additional melting.**

**This study is highly relevant since it addresses outstanding questions about the AMOC stability while considering the interaction between the AMOC and the Greenland Ice Sheet. At the same time, Greenland's meltwater release grows in a warming world. The authors highlight that it is not sufficient to consider a "fixed" threshold (overshoot tipping) beyond which the AMOC breaks down or GrIS disintegrates. They also underscore that changing rates could drive components in the coupled system beyond**

**stable conditions (rate-induced tipping). In addition, a stabilizing feedback, such as reduced North Atlantic warming due to a weakened AMOC, may not be strong enough to offset the disintegration of the GrIS. Here, the reduced model complexity allows scanning the phase space for numerous tipping cascades. Although, these are not necessarily representative of the natural world.**

**In general, it was a pleasure to read the well-structured manuscript. The figures are of high quality, necessary, and informative.**

**I recommend the publication of the manuscript after minor corrections.**

We are happy that the referee agrees on the relevance of this topic and are grateful for this positive evaluation of our manuscript and the figures. We thank the reviewer for carefully reading our manuscript and providing many helpful comments and explicit suggestions to improve our manuscript, in particular, with regard to the presentation of our results and clarifications of formulations.

**General comments**

**Although the manuscript is well organized and generally well written, section 4.2, including its subsections, could be better written. The language of section 4.2 is less clean than the remaining section. Therefore, I suggest revising this whole section.**

We have revised the language in Section 4.2, also following related and very helpful suggestions of Reviewer 3 (see below), and hope that it is clearer now. We would like to thank Reviewer 3 for the effort of creating detailed comments on our manuscript.

**Furthermore, the authors presented additional lengthy information in brackets, which may disturb the reader. I suggest the authors integrate this information into the general text or drop it if applicable.**

Following the comments from all reviewers, we have included all explanations in brackets either in the main sentence (separated by commas) or integrated them as additional sentences.

**In Figures three and four, small schematic icon-like figures of the remaining ice-thickness across Greenland seem to indicate remaining ice. If this is the case, please state more clearly where this pattern comes from because I do not see how the applied flowline ice sheet model can provide this pattern.**

We thank the reviewer for pointing out this inconsistency between the remaining ice thickness across Greenland in the icon-line figures and the possible outcomes provided by the applied flowline ice-sheet model. This remaining ice thickness was motivated by previous results with a more complex fully-dynamic ice-sheet model (Robinson et al. 2012). It is correct that such a spatial pattern cannot be produced by the flowline model used here. In the revised manuscript, we have modified the icon-line figures showing no remaining ice in Greenland for the ice-free state.

**Occupationally, terms/variables are introduced, which are defined later in the text. In some cases, this needs to be clarified. For example, the text refers (Page 9, Line 232) to Figure 2, where the variable appears in the caption, while it is later introduced on Page 9, Line 246.**

In the revised manuscript, we have made sure that all terms / variables are introduced before their use. In figure captions, abbreviations of variables are always paired with a descriptive term.

**Specific comments**

*Main document*

**Page 3, Line 79: Add comma in "... for a limited forcing, given that the …"**

Done.

**Page 4, Line 112: Please introduce the not SI-unit Sverdrup, for example, by a footnote or additional text.**

Done.

**Page 2, Line 36 – 37; Page 4, Line 107 – 198; Page 4, Line 109 – 111, Page 4, Line 118, Page 5, Line 1124 – 125: Missing citation (Madsen et al. 2022).**

We thank the reviewer for bringing this reference to our attention. We have included it in all relevant sections of the revised manuscript.

**Page 5, Line 132: Do you mean the citation "(compare e.g. Lohmann and Ditlevsen, 2021)?**

The given reference is the one that we intended to give. We agree that Lohmann and Ditlevsen (2021) fits as well, so we have added it in the revised manuscript.

**Page 5, Line 151: You may consider adding additional information: "(… continent by the ocean without floating ice shelves in Oerlemans (1981))."**

We have added this additional information in the revised manuscript.

**Page 6, Line 156: "Thereby, the surface mass balance $a_s$ that is the difference between mass gain and mass loss is reduced and …"**

We thank the reviewer for this suggestion and pointing out a missing explanation of the term 'surface mass balance' in the original manuscript. In the revised manuscript, we have added an explanation in the description of the Greenland Ice Sheet flowline model (lines 160-161):

> *The surface mass balance of an ice sheet is the sum of mass gain through precipitation and mass loss through runoff, erosion and sublimation runoff at its surface.*

**Page 6, Equation 3: Is the variable $\tilde{a_0}$ a spatial dependent variable? If so, please indicate or mention it. In addition, does this linear equation consider an increased vulnerability by a more than linear increase of the ablation zone for lowering height?**

We thank the reviewer for pointing out this lack of clarity in the original manuscript. $\tilde{a_0}$ is not spatially dependent. In the revised manuscript (line 176), we have added the following clarification:

*The surface mass balance for h = 0, $\tilde{a_0}$, is not spatially dependent.*

We agree that the use of this linear equation is a simplification in that respect. Including this effect by modifying the equation would likely accelerate the melting of the ice sheet on Greenland. Our modelling approach allows us to qualitatively study the dynamical regimes of the interacting Greenland Ice Sheet and AMOC. We do not aim to provide quantitative statements or projections on the emergence of tipping cascades in the climate system. While we do not take this effect into account here, it could be an interesting next step for future research.

**Page 6, Line 165: You may add at the end: "Furthermore, the surface mass balance equals in our setup the total mass balance."**

Done.

**Page 7, Equation 11: I can not find the definition of $A_i$ in the text. Please add it, even if this information is available in the supplement's table. Furthermore, does a hosing of H = 0.2 Sv, as shown in Figure 2b, correspond to an additional freshwater flux of $A_i \cdot H$ ? If so, wouldn't it correspond to a hosing freshwater flux of 0.014 Sv and 0.1504 Sv into the North Atlantic and Tropical Atlantic box, respectively? Since the manuscript addresses the impact of freshwater on the AMOC stability and the conditions at and around Greenland, it is surprising that the freshwater flux may be ten times larger in the tropics than in the North Atlantic. Anyhow, please clarify this point.**

We thank the reviewer for pointing out the missing definition of $A_i$ in the original manuscript.

The parameters $A_i$ in the global ocean box model are multiplicative factors for the hosing in the respective ocean boxes. That is, the 'uniform' hosing parameter H is scaled for each of the boxes, according to these factors. The box model parameters are calibrated based on GCM experiments (Wood et al., 2019). In this calibration, the values for the factors $A_i$ are chosen to match the total freshwater flux $A_i*H$ into each of the boxes as in the GCM when imposing hosing. Here, we have adopted the parameters obtained in a calibration with FAMOUS, following Wood et al. (2019).

An explanation is added in the revised manuscript (lines 218-220):

*Here, $F_{i0}$ are considered as baseline surface fluxes of the respective ocean boxes under pre-industrial conditions, and $A_i$ are multiplicative factors distributing additional surface freshwater fluxes across the boxes based on the hosing H (Wood et al., 2019).*

These parameters $A_i$ aim to reflect freshwater fluxes in a hosing experiment with FAMOUS and are here taken from Wood et al. (2019), Table 1. Given the parameter values $A_N = 0.07$ and $A_T = 0.752$ for the multiplicative factors in the North Atlantic and the Tropical Atlantic box, it is correct that the additional freshwater flux through hosing, e.g. with $H = 0.2$ Sv, in this parameter setting is larger for the Tropical Atlantic than for the North Atlantic box.

Combined with the 'baseline' surface freshwater fluxes $F_{i0}$ (inferred from FAMOUS under pre-industrial conditions), where $F_{N0} = 0.384$ Sv and $F_{T0} = -0.723$ Sv, the sum of these surface freshwater fluxes (compare Eq.11) is positive (that is, freshwater input) for the North Atlantic box for the considered range of hosing H. In this range of hosing H, it is negative (that is, freshwater loss) for the Tropical Atlantic box. In addition, the North Atlantic box receives meltwater from Greenland in our experiments, derived from the flowline model. We here would also like to refer to our response to the general comment related to the hosing by Reviewer 1.

**Page 8, Line 217: Does the surface mass balance on the ground correspond to the surface mass balance at the sea-level?**

This is a good question, in particular, in the context of the conceptual flowline setup used here to capture the tipping behaviour of the Greenland Ice Sheet. The surface mass balance on the ground refers to the surface mass balance where the ice thickness (and surface elevation) equals 0 in our model. The model capturing the Greenland Ice Sheet dynamics does not include any dependence on the sea-level changes. Given that the Greenland Ice Sheet predominantly rests on bedrock above present-day sea level, locations where the ice thickness equals 0 are above but may be close to sea level.

**Page 8, Line 218 – 219: You may rephrase "For a declining overturning strength $q$ of the AMOC with $H > H_{ref}$, the temperature $T_N$ in the North Atlantic box declines as well according to Eq. (4)" if applicable, e.g., "For active hosing ($$H > H_{ref}$), the AMOC overturning strength $q$ declines as well as the temperature $T_N$ in the North Atlantic box is driven by Eq. (4)."**

We have rephrased this sentence in the revised manuscript, along the lines of the reviewer's suggestion. In the revised manuscript (lines 240-242), this sentence now reads as:

*With $H > H_{ref}$, the AMOC overturning strength q declines. Driven by Eq. (4), the temperature $T_N$ in the North Atlantic box then declines as well.*

**Page 8, Line 219: You may rephrase "For $d_{oa} = 0$, we recover a …" with "For $d_{oa} = 0$, we obtain a …"**

Done.

**Page 8, Line 220: I find the wording "independent ice sheet" confusing and misleading. Please change it.**

We have included the reviewer's suggestion that is given in the following comment – the wording 'independent ice sheet' is thus removed in the revised manuscript.

**Page 9, Line 219 – 223: You may simplify these sentences: "A unidirectional coupling is obtained by $d_{oa} = 0$, where Greenland is not exposed to any changes in the North Atlantic (Eq. 12). In addition, the freshwater flux by the Greenland Ice Sheet resulting from its mass loss (Bamber et al., 2012, 2018; Trusel et al., 2018) is added as $F_{GIS}$ to the combined freshwater into the surface North Atlantic box as:"**

Done.

**Page 9, Line 229: Wouldn't it be correct to state "($F_{GIS} >0 {\mathrm Sv}$)"?**

Yes, thanks for pointing this out. This is adjusted in the revised manuscript.

**Page 9, Line 231: You may write "… freshwater flux into the Atlantic Ocean, which increases the …" or?**

Done.

**Page, Line 245 – 247: The sentence "More specifically, the surface …. is reached (Fig. 1(b))" is not clear. Please rephrase.**

Motivated by the reviewer's suggestion of changing the order of this sentence (see following comment), we have changed the sentence in the revised manuscript. In the revised manuscript (lines 278-281), it now reads as:

> *More specifically, the surface mass balance at the ground level $a_0$ is decreased linearly with a ramping rate $r_{a0}$ towards or across the deglaciation threshold $a_{0dgc}$. Once this deglaciation threshold is crossed, a stable ice sheet cannot be sustained. The surface mass balance on the ground is then kept constant after a final value $a_{0dgc} <= a_{0max}$ is reached (Fig. 1(b)).*

**Page 9, Line 245 – 247: You may change the order in this sentence: "More specifically, with a ramping $r_{ao}$, the ground surface mass balance $a_o$ decreases linearly and, once crossed the deglaction threshold $a_{0_{dgc}}$, the ice sheet stability is not sustainable." or "More specifically, with a ramping $r_{ao}$, the ground surface mass balance $a_o$ decreases linearly, and the ice sheet becomes unstable, once crossed the deglaction threshold $a_{0_{dgc}}$ is crossed.**

Please see our response to the previous reviewer's comment.

**Page 10, Line 256: You may rephrase "By decreasing the surface mass balance at the ground level associated … ."**

We have rephrased 'surface mass balance at the ground' to 'surface mass balance at the ground level' in the revised manuscript.

**Page 10, Line 256: You may rephrase to "… threshold and eventually disintegrates completely … ."**

Done.

**Page 10, Line 257: You may replace the beginning of the sentence: "In the following, AMOC hosing is kept constant."**

This sentence is removed in the revised manuscript following a comment by Reviewer 2.

**Page 10, Line 260 – 265: You may rewrite it: "The freshwater volume loss resulting from the forced deglaciation of Greenland corresponds to a time-varying GIS freshwater flux $F_{GIS}$ into the North Atlantic. This time-dependent GIS freshwater flux first increases as the GIS disintegrates. Consequently, the AMOC grows, potentially overshooting its threshold (Ritchie et al., 2021), but eventually returns to $F_{GIS} = 0 \mathrm{Sv}$ under otherwise constant hosing (Fig. 2(a), the AMOC trajectory approximately follows the black lines)".**

Done.

**Page 10, Line 267: Please consider replacing "observed" with "detected" when describing the results of simulations since model results are not measured and turned into observed properties. Therefore, please rephrase "… to a freshwater flux as detected in previous hosing experiments … ."**

We thank the reviewer for suggesting this much better wording. We have changed it in the revised manuscript.

**Page 10, Line 269 – 272: You may rephrase " In particular, the AMOC may transition to its 'off'–state in response to the Greenland Ice Sheet disintegration, which is accompanied by a temporary overshoot of the GIS freshwater flux threshold, resulting in an overshoot cascade (Fig. 2(c)). The increasing GIS freshwater flux puts the AMOC from the 'on' to the 'off'-state, while the AMOC does not recover after the decline of the GIS freshwater flux."**

We have rephrased this sentence in the revised manuscript, along the lines of the reviewer's suggestion. In the revised manuscript (lines 307-311), this sentence now reads as:

> *In particular, the AMOC may transition to its 'off'-state in response to the disintegration of the Greenland Ice Sheet. This may be accompanied by a temporary overshoot of the GIS freshwater flux threshold, resulting in an overshoot / bifurcation cascade (Fig. 2(c)). The increasing GIS freshwater flux takes the AMOC out of the basin of attraction of the 'on'-state, while the AMOC does not recover after the decline of the GIS freshwater flux with the deglaciation of Greenland.*

**Page 10, Line 272 – 273: The following might be more appropriate: "The surface mass balance decreases substantially… , which results in a complete deglaciation of Greenland."**

Done.

**Page 10, Line 276: I'm unsure, but shouldn't it be: "AMOC weakening without tipping, as commonly detected in hosing … "?**

Done.

**Page 10, Line 278 – 279: Here is an example of avoiding unnecessary brackets: "… within 1000 years driven by a faster and stronger … ."**

Done. Following the comments from the other reviewers, we have revised the whole manuscript and removed unnecessary brackets here (and elsewhere), whenever applicable.

**Page 10, Line 281: What do you think about rephrasing: "...deglaciation of Greenland, the AMOC leaves the stable 'on'-state. Rate-induced transition … ."?**

With this formulation, we aim to stress the differences between the intrinsic AMOC response timescale and the fast forcing (arising from the relatively fast GIS deglaciation), that are underlying this rate-induced transition.

The potential rate-dependence of tipping can be illustrated by the pulling of a tablecloth that lies underneath some dishes on a table (as introduced in e.g. Kiers, 2020). There are two distinct outcomes:

1)  The dishes come with the tablecloth, when slowly pulling the tablecloth. That is, the AMOC can keep up with and remains in the 'on'-state.
2)  The dishes are left behind on the table, when pulling fast. That is, the AMOC cannot keep up with the and leaves the 'on'-state.

In the revised manuscript (lines 325-326), we have rephrased this sentence along the lines of the reviewer's suggestion as follows:

> *With the relatively fast deglaciation of Greenland, the AMOC cannot keep up with the stable 'on'-state, leaves the stable 'on'-state and then crosses the moving basin boundary.*

**Page 10, Line 284: I suggest replacing the text with "the AMOC to the changing freshwater flux by … ."**

We thank the reviewer for this suggestion. In the revised manuscript, we would like to keep the phrase 'rate of change' to stress that an AMOC collapse may not only be triggered by crossing a critical value in the forcing, but also if the rate of change of this forcing is fast enough. We have added that this freshwater flux is then considered as time-dependent in the revised manuscript (lines 327-329), and hope that this sentence is now easier to understand:

> *More recently, Lohmann and Ditlevsen (2021) confirmed the suggested sensitivity of the AMOC to the rate of change of a time–dependent freshwater flux by demonstrating rate–induced tipping in a complex ocean model.*

**Page 10, Line 284: Please add a comma after the preposition: "Here, it is assumed … ."**

Done.

**Page 10, Line 285: Enclose the example by commas: " disturbances, e.g. in initial box salinities, are always .. ."**

Done.

**Page 10, Line 287: Add a comma for the subordinate clause at the end: "decline as studied, e.g. as scenario-dependent … ."**

Done.

**Page 12, Line 291 – 293: The end of the sentence is unclear; please improve the text.**

After identifying the different cascading dynamics arising for the Greenland Ice Sheet and the AMOC, we assess the occurrence of these different dynamic regimes in the parameter space. Here, the parameter space consists of the ramping rate $r_{a0}$ and the final value $a_{0max}$ of the surface mass balance on the ground level, determining the deglaciation time of the ice sheet on Greenland. Lines 291-294 of the original manuscript aims at introducing this next step in our analysis.

We have reformulated the sentence in the revised manuscript, and hope that it is clarified. In the revised manuscript (lines 344-347), this sentence now reads as:

> *By varying the ramping rate $r_{a0}$ and the final value $a_{0max}$ of the GIS surface mass balance on the ground, we systematically explore and quantify the occurrence of these different dynamic regimes; that is, the overshoot / bifurcation cascade and the rate-induced cascade of the Greenland Ice Sheet and AMOC.*

**Page 12, Line 294 – 295: Please extend the text to read: "of the tipping element drivers in our model."**

Done.

**Page 12, Line 309: add missing comma around "thus": " lower hosing values and, thus, for the AMOC … ."**

Done.

**Page 12, Line 312: I'm unsure, but I guess a comma is missing: "with a slow ice sheet decline, a high hosing determining the fixed … ."**

We have added a comma in the revised manuscript.

**Page 12, Line 314: The sentence needs to be clarified, or?**

We agree that the wording in this sentence is not entirely clear. We here aim at stating the conditions for a propagation of tipping from the Greenland Ice Sheet to the AMOC in an

overshoot / bifurcation cascade. In the case of a slower deglaciation, a larger hosing is necessary. The larger hosing brings the AMOC closer to the hosing threshold in the applied model.

We have extended this sentence in the revised manuscript, and hope that it is clarified by these additions. In the revised manuscript (lines 364-365), this sentence now reads as:

> *In other words, the AMOC has to be shifted closer to its hosing tipping point by increasing the hosing H for a propagation of tipping from the Greenland Ice Sheet to the AMOC.*

**Page 12, Line 315 – 316: I would like to suggest: "an overshoot cascade changes by variations of Greenland Ice Sheet's melting patterns. More … "**

Done. In combination with additional reviewer comments, this sentence (lines 366-368 in the revised manuscript) has been changed to:

> *The relative size of the region in the parameter space which gives rise to an overshoot / bifurcation cascade changes by variations of the Greenland Ice Sheet's disintegration time. More…*

**Page 13, Line 340 – 342: I suggest: Here, …. of an AMOC weakening, and it may be …. surface mass balance, for a warming … ."**

This sentence has been reformulated, also following a comment by Reviewer 2.

In particular, a more detailed explanation of effective tipping points for interacting tipping elements has been added in the revised manuscript (lines 389-403):

> *Considering this negative feedback, the intrinsic tipping point of the Greenland Ice Sheet (that is, the critical threshold of the Greenland Ice Sheet without any coupling, compare Fig. 4(c) and (f), dashed grey; Klose et al. 2020) is replaced by two separate effective deglaciation thresholds $a_{0dgc}^{(1)}$ and $a_{0dgc}^{(2)}$ of the GIS (Fig. 4(c) and (f), solid black), depending on the state of the AMOC. This is based on the theoretical foundations of cascading dynamics for linearly coupled driving (or 'master') and following tipping elements, formulated in Klose et al. (2020): Interactions shift the critical threshold of a responding system beyond which tipping is expected to lower or higher values compared to the intrinsic tipping point depending on the direction of coupling and the state of the driving tipping element, giving rise to effective tipping point(s) of the responding system. Here, when considering the stabilizing effect of an AMOC weakening on the ice sheet (Eq. (12)), the AMOC could be considered as the driving system, while the ice sheet on Greenland would represent the responding system. Based on Eq. (12), that linearly relates the AMOC state in terms of the North Atlantic box temperature and the GIS surface mass balance, two deglaciation thresholds $a_{0dgc}^{(1)}$ and $a_{0dgc}^{(2)}$ may then be crossed with a decreasing surface mass balance in a warming climate (Fettweis et al. 2013): For $a_0 < a_{0dgc}^{(1)}$ a complete melting of the GIS is obtained given that the AMOC resides and remains in its 'on'--state. Given that the AMOC resides in its 'off'--state, the GIS melts down completely for $a_0 < a_{0dgc}^{(2)}$.*

**Page 13, Line 344: Do you mean "distinct tipping thresholds" or "different tipping thresholds"?**

We thank the reviewer for asking for clarification. We here refer to the two effective deglaciation thresholds for the Greenland Ice Sheet. These arise and are relevant when considering a negative feedback from a relative cooling around Greenland with a weakened AMOC compared to the intrinsic deglaciation threshold. In that respect, 'distinct' and 'different' may be correct here.

To avoid confusion, we replaced 'distinct' by 'separate' in the revised manuscript (line 403).

**Page 13, Line 349: Please replace "observed" with "detected."**

Done.

**Page 15, Line 356 – 358: The sentence "With the AMOC tipping …. from the AMOC overturning strength" is unclear to me.**

This paragraph explains how the tipping of the Greenland Ice Sheet may be avoided by the stabilizing 'negative' part of the feedback loop. Here, the relative cooling in the North Atlantic with a weakening or tipping of the AMOC is relevant. In the modelling approach, this process is described by the linear dependence of the North Atlantic box temperature on the AMOC overturning strength, compare Eq. (4).

We have reformulated this sentence and included a reference to the relevant Eq. (4) in the revised manuscript. In the revised manuscript (lines 417-419), it now reads as:

> *With this AMOC tipping, a relative cooling of the North Atlantic box follows, given the assumed linear dependence of the North Atlantic box temperature on the AMOC overturning strength (Eq. (4)).*

**Page 15, Line 378: You may replace "ice sheet melting time" with "ice sheet disintegration time"?**

Thank you for this suggestion. We have changed the term in the revised manuscript.

**Page 17, Line 389: I guess a comma is missing: "is accelerating (Shepherd et al., 2020), and its … ."**

A comma is added in the revised manuscript.

**Page 17, Line 398: Since you are apparently using the British syntax predominately, replace "e.g.," with "e.g.".**

Done.

**Page 17, Line 404 – 406: Unclear sentence. Please rephrase.**

In this sentence, we aim at giving available evidence for changes in the AMOC that have already been observed. This includes the possibility that the AMOC gets closer to a critical threshold, though we would like to stress that evidence is limited here for now (compare our response to previous related comments).

We have rephrased this sentence in the revised manuscript, and hope that it got clearer. In the revised manuscript (lines 467-471), this sentence now reads as:

> *A decline of 15 % in the strength of the overturning circulation since the mid-twentieth century is found in the observed sea-surface temperature trend (Caesar et al., 2018) and it is suggested that the current AMOC state might lose stability (Boers, 2021; Van Westen et al., 2023).*

**Figure 1: Please increase the size of the hardly seen green points, which is stated in the text (Page 9, Line 235): "subcritical Hopf bifurcation at $F_{GIS}$ $Hopf (indicated by green points in Fig. 2(a))."**

We have increased the size of the green points in Figure 2 in the revised manuscript.

**Figure 1, caption: The introduced variable $r_{a0}$ has to be defined. Please find a way to introduce it and/or refer to the text.**

We agree that the definition of the parameters H, $r_{a0}$ and $a_{0max}$ could be improved. The evolution of the surface mass balance on the ground $a_0$, determined by the parameters $r_{a0}$ and $a_{0max}$ is described in lines 245-248 in the original manuscript. In the revised manuscript, we have adjusted the introduction of these parameters. In particular, we have moved this paragraph to introduce Section 4.2 on 'Tipping cascades between GIS and AMOC without negative feedback' in the revised manuscript. A reference to the text has been added in the figure caption.

**Figure 3, caption: I would like to suggest the following modification to the figure caption:" Shown is the AMOC overturning strength, also taking … " (drop "now"); "… declining from pink (100 %) to grey (0 %) as indicated by the right colorbar."; "indicate the AMOC in its 'on'-state, see bottom colorbar)."**

We thank the reviewer for the suggestion of referring to the respective colorbar, and have adjusted the caption of Figure 4 in the revised manuscript. In the revised manuscript the corresponding part of the section now reads as:

> *(a)-(b) & (c)-(d): Dynamics of the AMOC in terms of the overturning strength q over time. In addition, the GIS state in terms of the percentage of the initial GIS ice volume is shown in terms of the colouring declining from pink (100%) to grey (0%), compare colorbar on the right. The black lines indicate the 'on'- and the 'off'-state of the AMOC for the respective constant hosing without an additional freshwater input from Greenland ($F_{GIS}$ =0 Sv). (c) & (f): Tipping outcomes of GIS and AMOC for pathways of surface mass balance decrease with distinct constant hosing H within the ($a_0$,H)-plane. The respective tipping outcome is indicated by the colouring (grey: GIS deglaciation, pink: no GIS deglaciation; stripes additionally indicate the AMOC in its 'on'--state;*

*compare colorbar at the bottom of the figure). Solid black lines indicate the critical thresholds of the GIS and the AMOC. The intrinsic thresholds $a_{0dgc}$, which arises by neglecting the coupling via the temperature with a coupling strength $d_{oa}$ =0, is indicated as grey dashed lines.*

**Figure 3: Please define the green arrows in the caption and drop them.**

The green line in Figure 3 indicates all values of the hosing H for which the AMOC 'on'-state loses stability when additional mass loss and freshwater fluxes from Greenland is taken into account (compare figure caption). The green arrow was included as additional visualization. We agree that the green arrow, also in combination with a missing explanation and the use of green arrows in Figure 4, is confusing. In the revised manuscript, we have removed the green arrow in Figure 3.

*Supplement Material*

**Table S1: Please add missing units, e.g., "psu" for $S\_0$.**

We thank the reviewer for pointing out the missing units in Table S1. We have added units in Table S1 in the revised Supplementary Material, where applicable.

**Table S1, caption: You may also define the unit Sverdrup in the caption.**

We have defined the unit Sverdrup in the caption of Table S1 in the revised Supplementary Material.

**Table S2: What are the missing units of the listed salinity contents? Please add.**

We thank the reviewer for pointing out the missing units in Table S2. We have added units of the salinity contents in Table S2 in the revised Supplementary Material.

**Table S3: Since the hosing flux $H$ has the unit "Sv" in your figures (e.g., 2b), and the combined freshwater flux according to equation (11) shall result in "Sv" as well, the unit of the parameters $A\_i$ are dubious. Please check.**

The parameters $A_i$ are multiplicative factors, and are unitless (Wood et al., 2019). We have corrected the units in Table S3 in the revised Supplementary Material. In addition, we have added an explanation of the parameters $A_i$ in the revised manuscript, following previous comments of this and other reviewers.

[revised manuscript text omitted]